# Transitive Representation Learning Enhances Histopathology Annotation

Moritz Schaefer [1 2]   Zoe Piran [1 3]   Nils Philipp Walter [4]   Animesh Awasthi [5 6]   Christoph Bock [5 6]   Jure Leskovec [1]
Zinaida Good [2]

## Abstract

The characterization of histopathology with AI promises to assist clinical decision-making, but it is currently limited due to coarse-grained annotations that miss cellular identities. To overcome this gap, we bridge histopathological images, gene expression profiles, and natural-language descriptions using SPATIALWHISPERER, a tri-modal contrastive learning model. Our training integrates community-scale datasets comprising spatially resolved gene expression profiles paired with histopathology images, as well as single-cell gene expression profiles with detailed annotations. The shared gene expression modality implies a transitive relationship between images and textual annotations, which our method leverages to enable accurate zero-shot cell type annotation directly from H&E images. SPATIALWHISPERER outperforms published baselines, achieving relative AUROC gains of up to 15.9% across three benchmarks spanning 19 tissues and 20 cell types. When training with data from all three modality pairs, we observe performance gains in low-data regimes. We formalize our approach and present a sufficient condition under which this transitive alignment is induced. Our work establishes *transitive representation learning* for fine-grained interpretation of histopathology images.

---

[1]Department of Computer Science, Stanford University, Stanford, CA, USA [2]Department of Medicine, Stanford University, Stanford, CA, USA [3]Research and Early Development, Genentech, Inc., South San Francisco, CA, USA [4]CISPA Helmholtz Center for Information Security, Saarbrücken, Germany (work carried out at Stanford University) [5]Medical University of Vienna, Institute of Artificial Intelligence, Center for Medical Data Science, Vienna, Austria [6]CeMM Research Center for Molecular Medicine of the Austrian Academy of Sciences, Vienna, Austria. Correspondence to: Moritz Schaefer <moritzs@cs.stanford.edu>, Christoph Bock <christoph.bock@meduniwien.ac.at>, Jure Leskovec <jure@cs.stanford.edu>, Zinaida Good <zinaida@stanford.edu>.

*Proceedings of the 43rd International Conference on Machine Learning*, Seoul, South Korea. PMLR 306, 2026. Copyright 2026 by the author(s).

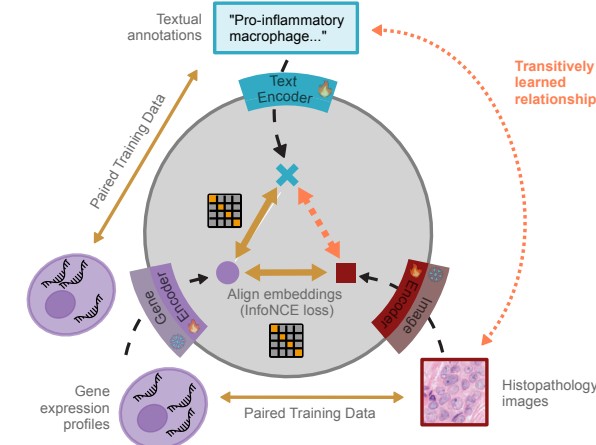

**Approach: Transitive Representation Learning**

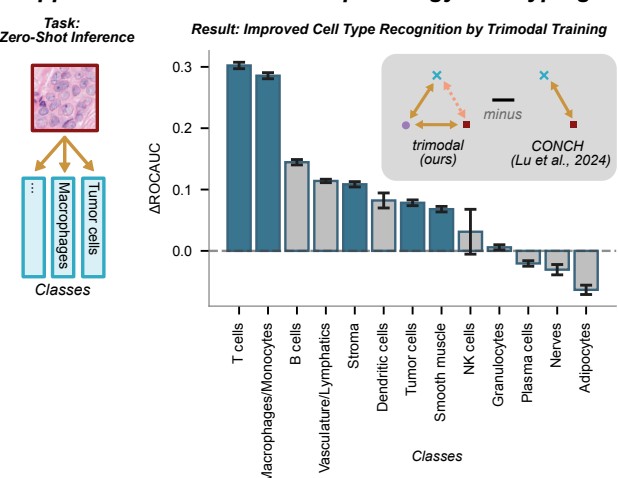

**Application: Zero Shot Histopathology Cell Typing**

*Figure 1.* **Top:** Overview of our trimodal SPATIALWHISPERER model. We train on *disjoint* paired datasets covering the edges *image ↔ gene expression* and *gene expression ↔ text*, which transitively induces an alignment between *image ↔ text*. **Bottom:** SPATIALWHISPERER enables zero-shot cell type prediction in histopathology image patches using arbitrary cell labels. Shown is the per-class $AUROC$ difference between SPATIALWHISPERER and a gold-standard pathology vision-language model (Lu et al., 2024). Error bars are 95% confidence intervals (over $n = 109$ sample images); blue bars mark classes with significant ($p < 0.05$) per-image differences. See Table 1 for the broader benchmark covering multiple datasets, baselines, and training seeds.

# 1. Introduction

Histopathology is a cornerstone of clinical diagnosis and biomedical discovery: stained tissue biopsies are imaged under a microscope and examined to detect cancer and characterize inflammatory and degenerative disease. Recent pathology foundation models and vision-language models, trained on large corpora of images and accompanying reports, promise to support this workflow through scalable retrieval, summarization, and zero-shot recognition of tissue patterns (Huang et al., 2023; Lu et al., 2024; Xiang et al., 2025). A central bottleneck in training these models is the limited availability of high-quality ground-truth annotations, because most labels and captions are assigned to large regions that contain hundreds to thousands of cells (Huang et al., 2023; Ikezogwo et al., 2023). More fine-grained annotations could enable the study of cellular composition and interaction in routine histopathology images.

Spatial gene expression profiling enables cell-level annotation and can be leveraged to partially infer gene expression from the more economical H&E histopathology data (He et al., 2020). To enable cellular interpretation of H&E images, recent work has attempted to derive cell-type annotations from inferred gene expression profiles (Chen et al., 2025a; Schaefer et al., 2025a). However, these approaches neglect the noise inherent in intermediate predictions (Wang et al., 2025), which impacts their performance.

In this work, we demonstrate how multimodal contrastive learning enables information transfer from richly annotated single-cell transcriptomics repositories to histopathology images, leveraging spatial transcriptomics as a cross-modal bridge. Spatial transcriptomics captures gene expression ($\mathcal{G}$) and microscopy images ($\mathcal{I}$), providing paired $\mathcal{I} \leftrightarrow \mathcal{G}$ samples at near-single-cell resolution. Over the past decades, expert-level annotations for dissociated single-cell data accumulated, providing rich paired $\mathcal{G} \leftrightarrow \mathcal{T}$ data. These paired resources share the gene-expression modality, suggesting a transitive route to learning image $\mathcal{I} \leftrightarrow \mathcal{T}$ relationships via indirect supervision (Fig. 1).

The key question is: *How can we combine these modality-adjacent datasets such that $\mathcal{G}$-associated cell-level semantic information $\mathcal{T}$ gets aligned with $\mathcal{I}$?*

Here, we model this transfer through an InfoNCE-based (van den Oord et al., 2018) trimodal contrastive objective that jointly aligns $\mathcal{I} \leftrightarrow \mathcal{G}$ and $\mathcal{G} \leftrightarrow \mathcal{T}$ in a shared space. We coin this approach *transitive representation learning* and provide a theoretical analysis confirming the expected information transfer across modalities: As the margins separating contrastive positives from negatives improve on the $\mathcal{I} \leftrightarrow \mathcal{G}$ and $\mathcal{G} \leftrightarrow \mathcal{T}$ pairs, an upper bound on the unobserved $\mathcal{I} \leftrightarrow \mathcal{T}$ loss tightens.

Empirically, we train SPATIALWHISPERER, a trimodal em-bedding model, on large, disjoint paired datasets. SPATIALWHISPERER outperforms a diverse set of H&E annotation approaches—leading pathology vision-language models, a two-stage H&E→expression→cell-type pipeline trained on identical paired data, and a published image–gene-expression bridging method—across the three independent cell-type benchmarks PathoCell, Lizard, and PanNuke, with relative AUROC gains of 13.9%, 15.9%, and 13.9% respectively over the strongest published baseline (Table 1, Fig. 1).

**Contributions.**

- We formalize *transitive representation learning*, previously observed as an emergent phenomenon (Girdhar et al., 2023), into a theoretically grounded framework. We derive an InfoNCE-based upper bound on the contrastive loss for the unobserved modality pair that tightens with the margins on the observed pairs, and identify failure modes such as misalignment of the shared bridge modality.

- We demonstrate this framework's merit for the biomedical community by training SPATIALWHISPERER—jointly aligning histopathology images, gene expression, and natural-language cell-state annotations—demonstrating zero-shot cell type inference that outperforms the state of the art across independent cellular pathology benchmarks.

- We characterize our approach through parameter and dataset ablations, observing that (i) representation learning substantially benefits scenarios where task-matching paired data is scarce, and (ii) harmonizing dataset overlaps enhances information transfer.

We provide our model weights and the source code to reproduce our analyses at `https://github.com/zinagoodlab/spatialwhisperer`.

# 2. Related Work

Multimodal contrastive learning has enabled scalable alignment of heterogeneous data sources by learning a shared embedding space from paired observations (Jia et al., 2021; Radford et al., 2021). In biomedicine, this paradigm has been applied to connect histopathology with textual reports (Huang et al., 2023; Lu et al., 2024; Xiang et al., 2025) and with spatial transcriptomics (Chen et al., 2025a; He et al., 2020; Nonchev et al., 2025; Tran et al., 2026; Valanarasu et al., 2026; Wang et al., 2025; Xie et al., 2023; Zeng et al., 2022), and to interface single-cell RNA-seq with textual annotations for zero-shot cellular analysis (Heimberg et al., 2025; Schaefer et al., 2025b; Zhao et al., 2024). Our work builds on these advances and studies how the

integration of disjoint paired datasets with a joint modality can transfer information to unobserved paired modalities.

In the paragraphs below, we contextualize related work on the integration of more than two data modalities.

**Simultaneous Alignment of Comprehensive Measurements.** When all modalities co-occur for the same sample, a natural approach is to align them together jointly. Recent methods propose geometric regularization or multi-view matching objectives to improve cross-modal structure (Cicchetti et al., 2025; Piran et al., 2024). However, in cellular biology and clinical pathology, it is uncommon to observe images, gene expressions, and rich textual annotations simultaneously at scale.

In this work, we leverage paired data, which are more common than complete sets, and ask how to use them to infer missing relationships.

**Hub-based Alignment with a Single Shared Modality.** Complementary work has integrated more than two modalities using pairwise data, learning a shared embedding space with images as a shared hub modality (Girdhar et al., 2023). Their work demonstrates transitive representation learning with images serving as the bridge modality, demonstrating that training-unseen relationships can be learned.

We build on this concept in the context of cellular biology using gene expression as the modality bridge. We provide a theoretical analysis of the conditions and failure modes under which transitive learning aligns representations, and support our findings with empirical results across diverse benchmark tasks.

**Autoregressive imputation.** Tran et al. (2023) target a disjoint-dataset setting through autoregressive inference. A transformer model first reproduces tokens of the bridge modality from a source, then reconstructs the target modality from those pseudo-tokens, propagating class-level supervision through weakly paired modality combinations.

In contrast, our method operates on strongly paired data and asks how to align, rather than generate, across unseen modality pairs.

**Biomedical Frameworks.** Integration of more than two modalities has been explored in biomedical contexts. Wang et al. (2024) freeze unimodal backbones and learn transformations between diverse modalities, such as drugs and phenotypes, from a knowledge graph. Chen et al. (2025b) combine diverse patient-centric modalities using both contrastive learning and task-specific objectives and a "mixture-of-modality-experts" architecture.

In contrast to these works, we learn a focused embedding space of cellular states and assess the effect of transitive representation learning on downstream task performance.

## 3. Method

### 3.1. Trimodal Contrastive Learning Framework

We learn a shared embedding space in which any two of the three modalities, histopathology images ($\mathcal{I}$), gene expression profiles ($\mathcal{G}$), and natural language ($\mathcal{T}$), can be compared via cosine similarity. The shared space should support (i) cross-modal retrieval (e.g., retrieve gene expressions or captions most compatible with an image patch), and (ii) prompt-based, zero-shot prediction (e.g., score a patch against a vocabulary of cell types). We build on paired data contrastive learning (Radford et al., 2021) and use modality-specific encoders followed by lightweight projection heads that map into a common $d$-dimensional space (Zhai et al., 2022).

**Encoders:** Let $\phi_{\mathcal{I}}, \phi_{\mathcal{G}}, \phi_{\mathcal{T}}$ denote the modality-specific encoders. Given inputs $x_i^{\mathcal{I}}, x_i^{\mathcal{G}}, x_i^{\mathcal{T}}$, our model produces unit-normalized embeddings $z_i^{\mathcal{X}} \in \mathbb{R}^{2048}$ in a shared space: $z_i^{\mathcal{I}} := \phi_{\mathcal{I}}(x_i^{\mathcal{I}})$, $z_i^{\mathcal{T}} := \phi_{\mathcal{T}}(x_i^{\mathcal{T}})$, and $z_i^{\mathcal{G}} := \phi_{\mathcal{G}}(x_i^{\mathcal{G}})$. These encoders are implemented through modality-specific backbone models stacked with two-layer MLP projection heads. We chose the following pretrained backbones in this study, which also define the input format for each modality

- **Image encoder:** *UNI 2* (Chen et al., 2024),[1] a pretrained vision transformer for histopathology patches of size $224 \times 224$ pixels (typically comprising $\sim$10–50 cells per patch).

- **Text encoder:** *BioBERT*, which embeds short natural-language descriptions of cellular phenotypes and tissue features (Lee et al., 2020).

- **Gene expression encoder:** *Geneformer*, which represents gene expression profiles as ranked gene tokens (Theodoris et al., 2023).

We use the inner product $\langle \cdot, \cdot \rangle$ as similarity. Under unit normalization, this equals cosine similarity.

**Zero-shot scoring.** Once trained, the model supports zero-shot prediction by comparing embeddings across modalities. For example, to predict a cell type for an image patch, we encode the patch into $z^{\mathcal{I}}$ and encode a set of candidate cell types as text prompts $\{x_c^{\mathcal{T}}\}$ into $\{z_c^{\mathcal{T}}\}$. We then score each candidate by cosine similarity $\langle z^{\mathcal{I}}, z_c^{\mathcal{T}} \rangle$ and choose the highest-scoring label. Analogous retrieval is possible across all modality pairs.

---

[1]Checkpoint https://huggingface.co/MahmoodLab/UNI2-h.

## 3.2. Training

We optimize a composite contrastive objective that aligns each observed modality pair. Following (Schaefer et al., 2025b; Zhai et al., 2022), we freeze the image and gene expression backbones and train the text encoder and all projection heads, keeping computation manageable while adapting the shared space to the paired datasets (see Appendix F.1 for an ablation of this choice and an alternative bridge encoder). Let $\lambda_{\mathcal{I}\leftrightarrow\mathcal{T}} = \lambda_{\mathcal{I}\leftrightarrow\mathcal{G}} = \lambda_{\mathcal{T}\leftrightarrow\mathcal{G}} = 1.0$ be scalar weights (set to 1.0 if not indicated otherwise; ablation in Appendix F.2) and $\tau$ a learnable temperature. In this work, we define

$$\mathcal{L} = \lambda_{\mathcal{I}\leftrightarrow\mathcal{T}}\,\mathcal{L}_{\mathcal{I}\leftrightarrow\mathcal{T}} + \lambda_{\mathcal{I}\leftrightarrow\mathcal{G}}\,\mathcal{L}_{\mathcal{I}\leftrightarrow\mathcal{G}} + \lambda_{\mathcal{T}\leftrightarrow\mathcal{G}}\,\mathcal{L}_{\mathcal{T}\leftrightarrow\mathcal{G}}. \quad (1)$$

where $\lambda_{\mathcal{I}\leftrightarrow\mathcal{T}}\,\mathcal{L}_{\mathcal{I}\leftrightarrow\mathcal{T}}$ is relevant only for experiments in Section 4.3 and otherwise set to 0.

Each pairwise term $\mathcal{L}_{\mathcal{X}\leftrightarrow\mathcal{Y}}$ for $(\mathcal{X},\mathcal{Y}) \in \{(\mathcal{I},\mathcal{T}), (\mathcal{I},\mathcal{G}), (\mathcal{T},\mathcal{G})\}$ is an InfoNCE loss (van den Oord et al., 2018). For a positive pair $(z_{\mathcal{X}}, z_{\mathcal{Y}})$ in a minibatch, we treat the paired sample as the unique positive and all other samples in the batch as negatives:

$$\mathcal{L}_{\mathcal{X}\leftrightarrow\mathcal{Y}} = -\log \frac{\exp(\langle z^{\mathcal{X}}, z^{\mathcal{Y}}\rangle/\tau)}{\sum_k \exp(\langle z^{\mathcal{X}}, z_k^{m_k}\rangle/\tau)}. \quad (2)$$

We train across the different datasets in an interleaved manner, such that minibatches contain data points reflecting different modality pairs, providing a rich contrastive signal.

Training is performed with a cosine learning schedule with initial warm-up for the first 3% of training steps and a learning rate of $10^{-5}$. The training batch size is set to 512, and we train for 4 epochs unless indicated otherwise.

Throughout the paper, our canonical SPATIALWHISPERER model refers to the configuration trained on the two paired datasets $\mathcal{T}\leftrightarrow\mathcal{G}$ and $\mathcal{G}\leftrightarrow\mathcal{I}$ (i.e. the bridge configuration); experiments that also incorporate the third $\mathcal{I}\leftrightarrow\mathcal{T}$ edge are flagged as such. SPATIALWHISPERER is trained with seed=0, and we trained two more identical models with varying seeds (seed=1, seed=2) to quantify run-to-run variability.

## 3.3. Analysis of Transitive Representation Learning

Here, we study when and how aligning the observed modality pairs $\mathcal{I} \leftrightarrow \mathcal{G}$ and $\mathcal{G} \leftrightarrow \mathcal{T}$ induces good alignment for the unobserved pair $\mathcal{I} \leftrightarrow \mathcal{T}$. First, we show why training the encoders on the observed pairs also reduces the loss on the unobserved pairs. For tractability, we initially assume perfect overlaps in the shared modality and quantify encoder quality via a uniform margin between positive and negative pairs. We then relax the perfect-overlap assumption, since real datasets rarely coincide exactly in the shared modality, and we show that the bound degrades smoothly with the

mismatch ($O(\sqrt{\delta})$ additive slack) rather than breaking. Finally, we discuss implications for training across all three edges and support the analysis with empirical insights in Section 4.

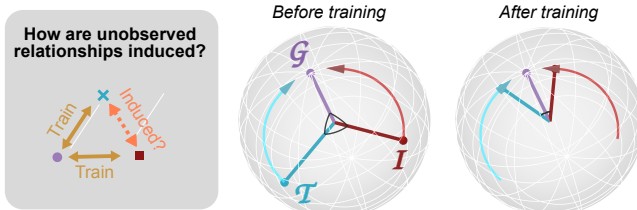

*Figure 2.* Intuition for transitive representation learning in a schematic 3-dimensional shared embedding space. As the InfoNCE loss shrinks the angular gaps on the observed pairs, the unobserved modalities become closer as well.

**Why transitive representation learning works.** Let $z_i^m$ denote the unit-normalized embedding of sample $i$ in modality $m \in \{\mathcal{I}, \mathcal{G}, \mathcal{T}\}$. As illustrated in Fig. 2, the composite objective intuitively aligns the unobserved pair by shrinking positive-vs-negative angular gaps on the observed pairs. In the positive case, if $z_{\mathcal{T}}$ is similar to $z_{\mathcal{G}}$ and $z_{\mathcal{G}}$ is similar to $z_{\mathcal{I}}$, then $z_{\mathcal{T}}$ and $z_{\mathcal{I}}$ will likely be similar as well. The caveat is that a low InfoNCE loss also depends on shrunken angular distances for the negative pairs, while those pairs are unobserved during training. Below we show why negative $z_{\mathcal{T}}, z_{\mathcal{I}}$ pairs are pushed apart in this setting such that effective transitive learning succeeds. Rather than deriving a strict guarantee of transfer under potentially unrealistic assumptions, we use a simplified setting to verify the intuition and guide the design of our experiments.

In contrastive learning, the quality of the encoder can be characterized by the separation margins $(\epsilon, \eta)$ it achieves between matched and mismatched pairs. More formally, for each sample $i$ and any distinct sample $j \neq i$, we require

$$\langle z_i^{\mathcal{I}}, z_i^{\mathcal{G}}\rangle \geq 1 - \epsilon, \qquad \langle z_i^{\mathcal{G}}, z_i^{\mathcal{T}}\rangle \geq 1 - \epsilon, \quad \text{(P)}$$

$$\langle z_i^{\mathcal{I}}, z_j^{\mathcal{G}}\rangle \leq \eta, \qquad \langle z_i^{\mathcal{G}}, z_j^{\mathcal{T}}\rangle \leq \eta. \quad \text{(N)}$$

Thus, true pairs are close (similarity at least $1 - \epsilon$), while mismatched pairs are bounded by $\eta$.

Under these margins, we bound the per-sample InfoNCE loss for the unobserved $\mathcal{I} \rightarrow \mathcal{T}$ query in terms of $(\epsilon, \eta)$, making the intuition for transitive learning precise and capturing a sufficient condition for when the transfer succeeds. Here, we use the gene expression encoder as a semantic reference that links the other two modalities. Note that this choice depends on the available data.

**Lemma 3.1** (Per-sample InfoNCE bound for $\mathcal{I} \rightarrow \mathcal{T}$)**.** *Given a query $x_i^{\mathcal{I}}$, its matched (but unobserved) $x_i^{\mathcal{T}}$, and $N$ un-*

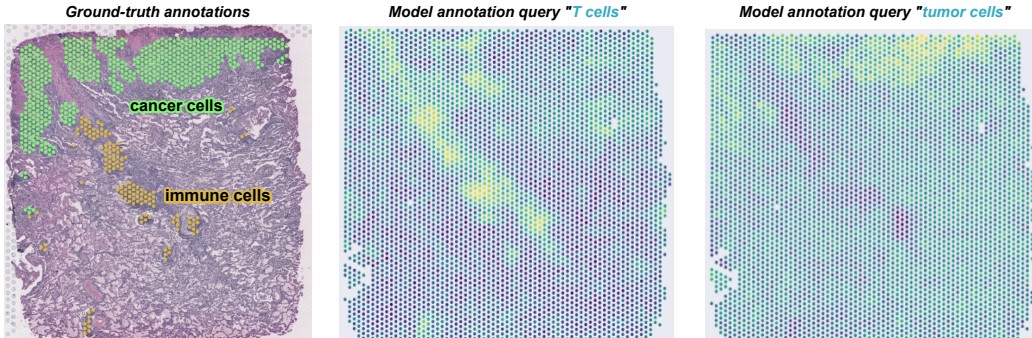

*Figure 3.* Zero-shot cell type prediction on a held-out lung cancer tissue sample using free-text queries. Inference is performed at the patch level, and colormaps show cosine similarities between encoded image patches and each of the two encoded queries. The shown sample was not part of our training data. Ground truth annotations were provided by an expert histopathologist. The original label for immune cells (yellow) was 'tertiary lymphoid structure', a niche characterized by dense immune cell infiltration, including T cells.

paired $x_{j_1}^{\mathcal{T}}, \dots, x_{j_N}^{\mathcal{T}}$ *contrastive negatives, define*

$$s^+ = \langle z_i^{\mathcal{I}}, z_i^{\mathcal{T}} \rangle, \qquad s_k^- = \langle z_i^{\mathcal{I}}, z_{j_k}^{\mathcal{T}} \rangle.$$

*Let $\tau > 0$ be the temperature. If* (P) *and* (N) *hold with margins $(\epsilon, \eta)$, then the loss $\ell_i$ across modalities $\mathcal{I}$ and $\mathcal{T}$ is bounded by $\epsilon$ and $\eta$:*

$$\ell_i = \log\Big(1 + \sum_{k=1}^{N} e^{(s_k^- - s^+)/\tau}\Big) \leq \log\Big(1 + N\, e^{r(\epsilon, \eta)/\tau}\Big),$$

*where $r(\epsilon, \eta) = q(\epsilon, \eta) - p(\epsilon)$ with $p(\epsilon) = 2(1-\epsilon)^2 - 1$ and $q(\epsilon, \eta) = \max\{\eta, (1-\epsilon)\eta\} + \sqrt{2\epsilon - \epsilon^2}$.*

Holding $\tau$ and $N$ fixed, the bound decreases as observed positives tighten ($\epsilon \downarrow$) and observed negatives are pushed away ($\eta \downarrow$). As $\epsilon \to 0$ and $\eta \to -1$, it approaches the optimal loss $\log(1 + Ne^{-2/\tau})$, which tends to 0 as $\tau \to 0$. Proof details are in Appendix A; conceptually, we decompose each embedding along the $\mathcal{G}$ *reference direction* and bound positive and negative projections and residuals.

We empirically probe these conditions in Section 4.

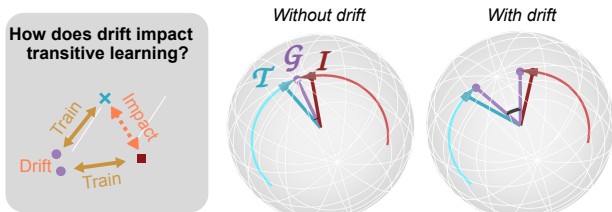

*Figure 4.* Impact of drift in the overlapping modality: imperfect overlap introduces an additive slack in the unobserved pair loss bound.

**Effect of imperfect overlap in $\mathcal{G}$.** Real datasets rarely coincide exactly in the shared bridge modality (Fig. 4). If the $\mathcal{G}$ reference directions differ across datasets, an extra change-of-reference projection introduces a slack in the exponent gap $r$. In the high-overlap regime $\langle z_i^{\mathcal{G}}, z_j^{\mathcal{G}} \rangle \geq 1 - \delta$, the degradation is an additive $O(\sqrt{\delta})$ (with smaller $O(\delta)$ and mixed $O(\sqrt{\epsilon\delta})$ terms), vanishing as $\delta \to 0$. The transitive bridge thus degrades predictably with bridge-modality mismatch rather than breaking.

We observe this empirically in Section 4.2: harmonizing the bridge-modality overlap across datasets tightens cross-modal alignment, in line with the predicted scaling. The result motivates data curation and diagnostics of overlapping modalities, such as distributional similarity checks.

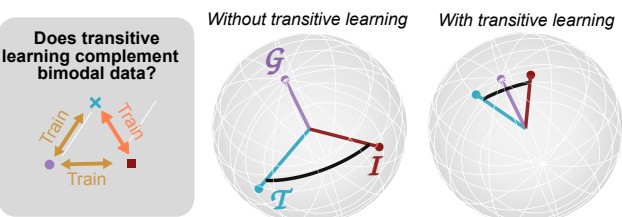

*Figure 5.* Leveraging transitive pretraining: transitive learning preconditions the unseen modality pair to be close, easing subsequent fine-tuning on the third edge. The black arc between $\mathcal{I} \leftrightarrow \mathcal{T}$ indicates angular distance to be refined by training.

**Implications for training on all three edges.** Under our trimodal objective, training can be performed jointly across datasets covering all three pairwise modality combinations. In this setting, each observed pair receives both direct supervision from its own contrastive loss and transitive supervision induced by the two remaining pairwise objectives. Our analysis illustrates how these two signals complement each other. Imagine this as a sequential learning protocol, where optimization on the transitive pairs $\mathcal{I} \leftrightarrow \mathcal{G}$ and $\mathcal{G} \leftrightarrow \mathcal{T}$ tightens an upper bound on the loss of the third pair $\mathcal{I} \to \mathcal{T}$, effectively pre-conditions the shared space. Including data for $\mathcal{I} \leftrightarrow \mathcal{T}$ then provides direct signal that further refines alignment. As a result, combined training re-

*Table 1.* Mean AUROC across three cell-type benchmarks (see benchmark preparation in Appendix B) for SPATIALWHISPERER and five baselines (described in Appendix C); per-class breakdown in Table 4. Best per column is **bold**; second-best is underlined. SPATIALWHISPERER performance is reported as the mean and standard deviation across three training seeds (0, 1, 2).

| Method | PathoCell | Lizard | PanNuke |
|---|---|---|---|
| **SPATIALWHISPERER** | **0.621 ± 0.008** | **0.749 ± 0.016** | 0.687 ± 0.001 |
| CONCH (Lu et al., 2024) (Appendix C.1) | 0.546 | 0.615 | 0.604 |
| PLIP (Huang et al., 2023) (Appendix C.1) | 0.488 | 0.528 | 0.584 |
| Bimodal $\mathcal{I}\leftrightarrow\mathcal{T}$ (QUILT-1M training data; Appendix C.2) | 0.566 | 0.642 | **0.713** |
| Two-stage UNI2→Geneformer (Appendix C.3) | 0.550 | 0.662 | 0.599 |
| OmiCLIP, short markers (Appendix C.4) | 0.491 | 0.611 | 0.520 |
| OmiCLIP, extended markers (Appendix C.4) | 0.478 | 0.646 | 0.525 |

quires less $\mathcal{I} \leftrightarrow \mathcal{T}$ supervision to achieve strong alignment (Fig. 5). We demonstrate this phenomenon empirically in Section 4.3.

## 4. Experiments

Our approach enables fine-grained analysis of histopathology images using free-text queries. We trained SPATIAL-WHISPERER on paired $\mathcal{I} \leftrightarrow \mathcal{G}$ and $\mathcal{G} \leftrightarrow \mathcal{T}$ datasets (see Table 2), withholding direct $\mathcal{I} \leftrightarrow \mathcal{T}$ supervision so that our evaluations isolate transitive learning of $\mathcal{I} \leftrightarrow \mathcal{T}$ mediated via gene expression.

Applying this model to annotate a lung cancer histopathology sample (Dawo et al., 2025; Schaefer et al., 2025a), produces ground-truth-matching annotations of cancer and immune cell populations using free-text cell-type labels as zero-shot queries (Fig. 3).

In this section, we systematically assess our trimodal approach through three complementary experiments that directly probe the analyses presented in Section 3.3. First, we train on two datasets and rigorously measure the model's transitively learned cell-type detection performance through a benchmark of annotated histopathology images. We compare this model against established and self-trained pathology vision-language models. Second, we test the impact of shared data modality overlap on information transfer. We harmonize annotation style in the text modalities across datasets and measure its effect on cross-modal retrieval. Third, we assess how transitive pretraining complements task-matched paired data. We examine this with a focus on training data scarcity, as high-quality annotations are often rare.

Unless stated otherwise, all evaluations are zero-shot: We query the learned shared embedding space with modality-specific encoders and perform nearest-neighbor retrieval or prompt-based scoring without task-specific fine-tuning.

**Datasets.** Table 2 summarizes the training and evaluation datasets used throughout this work. We leverage three large

bimodal datasets for training: HEST-1K ($\mathcal{I}\leftrightarrow\mathcal{G}$), CellWhisperer ($\mathcal{G}\leftrightarrow\mathcal{T}$), and, in selective experiments, QUILT-1M ($\mathcal{I}\leftrightarrow\mathcal{T}$). For compatibility with the $\mathcal{T}\leftrightarrow\mathcal{G}$ data, we restrict HEST-1K to Visium samples, which provide whole-genome expression profiles. The corresponding samples resolve close to the single-cell scale ($\sim$10 cells per patch). For QUILT-1M, which contains heterogeneous image sources, we keep only high-magnification histological content, as annotated by the authors. As evaluations are zero-shot or retrieval-based, no benchmark labels or data were seen during training. We evaluate on six datasets spanning cell-type prediction (PathoCell, Lizard, PanNuke) and each of the three modality pairs (Tabula Sapiens, HEST-1K, Skin Conditions) at patch-level granularity.

### 4.1. Transitive Representation Learning Improves Cell Type Prediction in Histopathology

To assess whether gene-expression-mediated transitive representation learning improves cell-level annotation in histopathology, we perform zero-shot cell type inference by scoring each image patch embedding against textual embeddings of candidate cell types in the shared embedding space. We started our performance benchmarks by comparing SPATIALWHISPERER against a leading pathology vision-language model (Lu et al., 2024) on the PathoCell benchmark of colorectal cancer samples (Fig. 1). SPATIAL-WHISPERER outperformed this baseline in 9 of 13 cell types, with the largest gains for immune and vasculature populations that are well captured by gene expression assays. Cell types with poor recognition performance coincide with known measurement limitations, as Adipocytes, for example, exhibit few molecules and are therefore poorly captured through gene expression profiling.

Applying the same evaluation across three independent cell-type benchmarks (PathoCell, Lizard, PanNuke; Appendix B) and six baselines spanning pathology VLMs, marker-gene bridging, and a two-stage image→expression→cell-type pipeline Appendix C), SPATIALWHISPERER leads in mean AUROC, with relative gains of 13.9%, 15.9%, and 13.9%

*Table 2.* Experimental setup: training datasets and evaluation benchmarks. Training datasets each provide paired data for two of the three modalities, covering the edges $\mathcal{I}\leftrightarrow\mathcal{G}$, $\mathcal{G}\leftrightarrow\mathcal{T}$, $\mathcal{I}\leftrightarrow\mathcal{T}$. Cell-type prediction benchmarks evaluate patch-level $\mathcal{I}\rightarrow\mathcal{T}$ classification and feed Table 1. Modality-pair benchmarks evaluate each training modality pair at sample-set granularity under low-$n$ subsampling (Fig. 6).

| Dataset | Modality / Task | Scope | Description |
| --- | --- | --- | --- |
| *Training datasets* | | | |
| HEST-1K (subset) (Jaume et al., 2024) | $\mathcal{I}\leftrightarrow\mathcal{G}$ | 921,154 pairs (384 samples) | Histopathology patches paired with spatially resolved Visium gene expression profiles. |
| CellWhisperer (Schaefer et al., 2025b) | $\mathcal{G}\leftrightarrow\mathcal{T}$ | 1,082,413 pairs | Gene expression profiles paired with textual annotations of cellular and tissue phenotypes; sourced from CELLxGENE Census (CZI Single-Cell Biology Program et al., 2023) and GEO/ARCHS[4] (Clough et al., 2024; Lachmann et al., 2018). |
| QUILT-1M (subset) (Ikezogwo et al., 2023) | $\mathcal{I}\leftrightarrow\mathcal{T}$ | 104,362 pairs | Histopathology images with matched captions, filtered to high-magnification histological content. Used in Section 4.3. |
| *Cell-type prediction benchmarks* (feed Table 1) | | | |
| PathoCell (Lüscher et al., 2025) | $\mathcal{I}\rightarrow\mathcal{T}$ zero-shot | 13 classes; 109 FoVs / 35 patients | Colorectal cancer tissue regions with cell-type labels, aggregated to patch-level presence. Details in Appendix B.1. |
| Lizard (Graham et al., 2021) | $\mathcal{I}\rightarrow\mathcal{T}$ zero-shot | 3 reduced classes; 70 colon adenocarcinoma datasets | Nucleus-level cell-type labels aggregated to patch-level presence; CRAG, GlaS, DigestPath subsets. Details in Appendix B.2. |
| PanNuke (Gamper et al., 2019) | $\mathcal{I}\rightarrow\mathcal{T}$ zero-shot | 4 reduced classes; 51 datasets / 19 organs | Nucleus-level labels across a broad organ panel aggregated to patch-level presence. Details in Appendix B.3. |
| *Modality-pair benchmarks* (feed Fig. 6) | | | |
| Tabula Sapiens (The Tabula Sapiens Consortium et al., 2022) | $\mathcal{G}\rightarrow\mathcal{T}$ zero-shot | 483,152 cells; 177 cell-type classes | Consortium-scale atlas of human cells with expert-curated annotations. Applied as in (Schaefer et al., 2025b). |
| HEST-1K Benchmark (Jaume et al., 2024) | $\mathcal{G}\leftrightarrow\mathcal{I}$ retrieval | 8 cancers; 9 organs | Official held-out test set partition for image$\leftrightarrow$expression retrieval (fully complementary to the training subset above). |
| Skin Conditions (Kriegsmann et al., 2022) | $\mathcal{I}\rightarrow\mathcal{T}$ zero-shot (Appendix B.4) | 16 conditions | Histopathology H&E patches of skin specimens with malignant and nonmalignant annotations. |

over the strongest published baselines on PathoCell, Lizard, and PanNuke, respectively (Table 1). Despite testing two representative marker-gene sets following the authors' pipeline, the OmiCLIP baseline recognized only B cells and fibroblasts at meaningful AUROC; other classes were near random, reinforcing that joint alignment of image, expression, and annotation outperforms a marker-gene-only bridge (per-class breakdown: Table 4).

We observe similar trends for other metrics and discuss those in Appendix D. We also evaluated alternative query formulations, which led us to use bare cell type labels as those performed the best in our baselines (see Appendix B).

### 4.2. Transitive Representation Learning Requires Overlapping Modalities

As highlighted in Section 3.3, transfer depends on an overlap in the shared modality. For example, as the text descriptions

associated with images and gene expressions deviate, the model may not be able to reliably link cell labels with image patches. In practice, this mismatch can be due to stylistic inconsistencies (e.g., different vocabularies, levels of specificity, and biological focus) even when the underlying semantics match.

To probe this effect, we curated the labels in the $\mathcal{I}\leftrightarrow\mathcal{T}$ QUILT-1M dataset to better align with the style of the labels in the $\mathcal{G}\leftrightarrow\mathcal{T}$ dataset. We used a large language model to rewrite the labels to mimic the annotation style of 20 hand-curated $\mathcal{G}\leftrightarrow\mathcal{T}$ samples that we use as in-context examples (see Appendix E). Then, we evaluated how well the curated versus the original labels aligned to their linked images under a model trained only on the other two modality pairs.

Consistent with our expectations, text harmonization yields significantly higher $\mathcal{I}\leftrightarrow\mathcal{T}$ similarity scores (Fig. 7), which

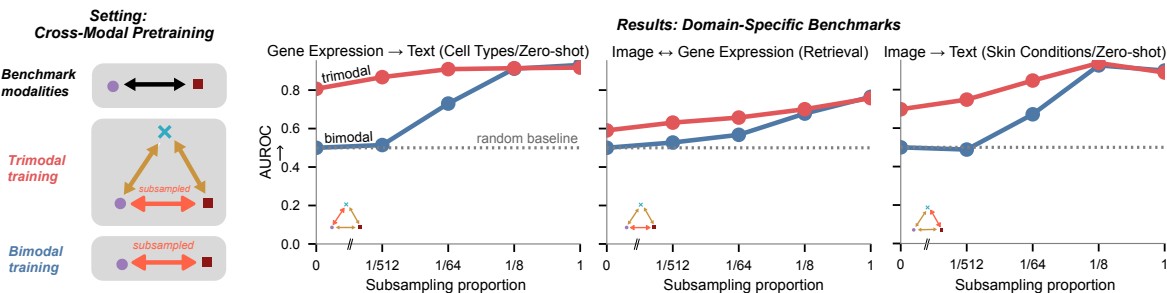

*Figure 6.* Effect of task-matching data with (red line; trimodal model) and without (blue line; bimodal baseline) transitive data. Task-matching data is provided in different amounts, to simulate low-data scenarios. Modality-matching tasks are described in Table 2. Scores are macro-averaged across classes. Baseline performance at x=0 is set to 0.5 (corresponding to fully random AUROC performance).

also reflects in better retrieval scores (AUROC 0.695 vs 0.645; computed on 20,000 sampled pairs for tractability). When including curated QUILT-1M data in trimodal model training, we observed considerable performance improvements over uncurated data inclusion for PathoCell cell-type prediction (mean 0.645 vs 0.609; Appendix E.2).

These results underline the importance of overlapping modalities for transitive representation learning. Targeted data curation can thus function as a practical lever for strengthening the alignment of unobserved modality pairs, both at the level of held-out $\mathcal{I} \leftrightarrow \mathcal{T}$ retrieval and when used as training data.

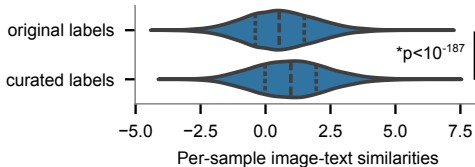

*Figure 7.* Comparison of cross-modal similarity for curated and original $\mathcal{I} \leftrightarrow \mathcal{T}$ data. Shown is the distribution of temperature-scaled cosine similarities across samples in (Ikezogwo et al., 2023) computed using a trimodal model trained with $\mathcal{I} \leftrightarrow \mathcal{G}$ and $\mathcal{G} \leftrightarrow \mathcal{T}$ data. Black lines indicate distribution quartiles. Statistics: Two-sided Mann-Whitney U

### 4.3. Task-Matching and Transitive Data Jointly Boost Performance in Low-$N$ Data Regimes

We finally asked whether combining task-matched and transitive data during training can improve model performance. We studied this question in a simulated scenario of Low-$N$ task-matched data, mimicking a common scenario in biomedical fields where high-quality labeled data is scarce.

For this purpose, we collected representative benchmark tasks covering each of the three modality pairs (Table 2, *Modality-pair benchmarks*). We then randomly subsampled the paired target data to fractions $\{1, 1/8, 1/64, 1/512\}$, along with a 0-data control (Fig. 6) and trained two sets

of models. First a bimodal model using only the subsampled task-matched data pairs, and second a trimodal model that additionally leverages the other two paired datasets for transitive representation learning through our trimodal objective.

Across all three benchmark tasks, incorporating auxiliary paired supervision via the shared modality, i.e., using our transitive representation learning approach, improved performance over the bimodal baseline, with the largest gains appearing in the smallest-data regimes (Fig. 6). At full provided data, the benefits vanish in most benchmarks, which we attribute to a combination of saturated benchmarks and complementary information in the transitive and task-matching data modalities. Beyond the cell-type benchmark, training with all three modality pairs also lifts coarser pathology tasks: PanNuke 19-class tissue-of-origin classification reaches AUROC 0.649, up from 0.586 for the transitive $\mathcal{T} \leftrightarrow \mathcal{G} + \mathcal{G} \leftrightarrow \mathcal{I}$ bridge (seed = 0; Appendix E.3).

## 5. Discussion

Gene expression profiling provides rich, mechanistically interpretable labels that can inject cellular semantics into histopathology. By training on paired $\mathcal{I} \leftrightarrow \mathcal{G}$ and $\mathcal{G} \leftrightarrow \mathcal{T}$ data, our model learns an implicit $\mathcal{I} \leftrightarrow \mathcal{T}$ mapping that enables zero-shot cellular annotation from routine H&E using natural-language queries. This establishes a practical route to semantic interpretation of cell and tissue morphology, beyond gene-level predictions.

Only a subset of transcriptional features can be derived at relevant signal-to-noise ratios from H&E morphology, a limitation consistently reported across image→expression benchmarks (Nonchev et al., 2025; Tran et al., 2026; Wang et al., 2025). This makes downstream interpretation challenging, as noisy gene expression is used in cell-level analyses that expect experiment-grade signals. We observe this shortcoming in published (Chen et al., 2025a) and newly trained baselines and address this challenge by jointly training a shared embedding across all three relevant modalities.

This leads to the targeted retrieval of recoverable H&E features that are relevant to downstream interpretation, such as the inference of cell types and tissue origin.

Transitive representation learning also improves data efficiency. In low-data regimes common in biomedical settings, where high-quality labels are costly, leveraging auxiliary paired datasets acts as a strong regularizer. Across 3 benchmarks, gains are most pronounced when target pairs are scarce, motivating a pretraining strategy in which broadly available paired corpora, for example spatial transcriptomics and atlas annotations, are used to strengthen task-specific bimodal performance in problems with low data availability.

Our analysis and experiments provide actionable insights into when transfer succeeds or fails. Most prominently, effective transfer depends on the overlap in the shared modality across datasets. When overlap is imperfect, the bound degrades predictably with the mismatch, which we observe empirically. We show this can be addressed through data preprocessing and leave other strategies such as confounder-aware encoder modules for future investigations. Simple diagnostics, such as empirical margin estimates and hub-overlap scores in the gene expression modality, can be computed during preparation for training to anticipate transitive potential and to guide curation.

Fine-grained cell-type annotation from routine H&E without task-specific labels has remained an open problem of computational pathology. Our framework couples the single-cell transcriptomics ecosystem to histopathology. We expect that multimodal foundation models for biomedicine will increasingly rely on compositional training setups, where large but incomplete paired datasets are combined to approximate richer supervision than any single dataset can provide.

### 5.1. Limitations & Future Work

While our results demonstrate that molecularly informed supervision can transfer effectively across modalities, several limitations point to clear opportunities for future work.

First, because our image inputs are patch-level, each example aggregates signals from multiple cells, which limits cell-resolved supervision. As spatial data at sub-cellular resolution is increasingly becoming available, we expect our approach to further improve trimodal model performance by aligning true single-cell data across all modalities.

Second, we chose state-of-the-art modality backbones and focused this work on a systematic comparison of training regimes. Future work can explore alternative encoders, parameter freezing, and adapter mechanisms to better cope with heterogeneity in the data (e.g., bulk, single-cell, and spatial transcriptomics) and potentially improve alignment.

Third, although we evaluate on established benchmarks covering representative modality pairs, the clinical and biological space is far broader. Expanding evaluation to additional tissues and tasks, and out-of-distribution testing across scanners and gene expression profiling assays will be necessary to foster wider data compatibility of our model.

Fourth, training data biases across tissues, demographics, and assays propagate into predictions and may amplify miscalibration on under-represented cell types and conditions.

Fifth, the bridge modality upper-bounds information transfer in both scope and alignment. Signals absent from gene expression, such as tissue-architectural features that benefit from direct image-text supervision, are unreachable through transcriptomics alone. This bottleneck is exacerbated by style or formatting mismatch across paired datasets, which can be partially recovered through data harmonization of the bridge modality (Section 4.2).

## Impact Statement

SPATIALWHISPERER advances multimodal alignment toward cell-resolved histopathology annotation. The released model has not undergone prospective clinical validation, pathologist-in-the-loop assessment, or regulatory clearance, and must not be used for diagnosis or treatment.

## Acknowledgements

We thank the whole Leskovec and Good labs for discussions and for providing feedback on our manuscript. This work was in part supported by the NIH/OD 1OT2OD038101 (Z.G., J.L.), Stanford Center for Digital Health Award (Z.G., J.L.), Laude Institute MOONSHOTS // ONE Honorable Mention Award (Z.G., J.L.), NIH/NCI 2P01CA049605 (Z.G.), NIH 1U24NS146314 (J.L.), Kona Innovation Challenge Award from the Parker Institute for Cancer Immunotherapy (PICI; C-04134), Weill Cancer Hub West PROMISE Award (Z.G.), NSF CCF-1918940 (Expeditions; J.L.), DMS-2327709 (IHBEM; J.L.), IIS-2403318 (III; J.L.), Stanford Data Science Applications Initiative (J.L.), Wu Tsai Neurosciences Institute (J.L.), Stanford Institute for Human-Centered AI (J.L.), Amazon (J.L.), Genentech (J.L.), SAP (J.L.), and SCBX (J.L.). Z.G. was supported by the Parker Bridge Award (PICI C-02895) and the NIH/NCI Pathway to Independence Award (1K99CA293149, 4R00CA293149). Moritz Schaefer is supported by a Cancer Research Institute Immuno-Informatics Postdoctoral Fellowship (CRI17992). The content is solely the responsibility of the authors and does not necessarily represent the official views of the funding entities.

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

# A. Proof & Discussion of the Lemma

**Intuition:** We declare the reference modality as a *reference direction* to decompose each embedding into its component along the reference and an orthogonal residual. The positive constraints make both projections large and the residual term small (by Cauchy–Schwarz), yielding a uniform lower bound on the positive score, while the negative constraint caps the negative projection and the same residual control yields a uniform upper bound on each negative score. Substituting these two bounds into the InfoNCE and upper-bounding the log-sum by $N$ times the worst-case gap gives the stated per-sample bound, which tightens as $\epsilon \downarrow$ and $\eta \downarrow$.

Analogously to modalities $\mathcal{I}, \mathcal{G}, \mathcal{T}$ in the main text, here we use $A, B, C$.

**Lemma 3.1** (Per-sample InfoNCE bound for $\mathcal{I} \to \mathcal{T}$)**.** *Given a query $x_i^{\mathcal{I}}$, its matched (but unobserved) $x_i^{\mathcal{T}}$, and $N$ unpaired $x_{j_1}^{\mathcal{T}}, \ldots, x_{j_N}^{\mathcal{T}}$ contrastive negatives, define*

$$s^+ = \langle z_i^{\mathcal{I}}, z_i^{\mathcal{T}} \rangle, \qquad s_k^- = \langle z_i^{\mathcal{I}}, z_{j_k}^{\mathcal{T}} \rangle.$$

*Let $\tau > 0$ be the temperature. If* (P) *and* (N) *hold with margins $(\epsilon, \eta)$, then the loss $\ell_i$ across modalities $\mathcal{I}$ and $\mathcal{T}$ is bounded by $\epsilon$ and $\eta$:*

$$\ell_i = \log\left(1 + \sum_{k=1}^{N} e^{(s_k^- - s^+)/\tau}\right) \leq \log\left(1 + N\, e^{r(\epsilon, \eta)/\tau}\right),$$

*where $r(\epsilon, \eta) = q(\epsilon, \eta) - p(\epsilon)$ with $p(\epsilon) = 2(1 - \epsilon)^2 - 1$ and $q(\epsilon, \eta) = \max\{\eta, (1 - \epsilon)\eta\} + \sqrt{2\epsilon - \epsilon^2}$.*

*Proof. Proof idea.* To bound $\ell_i$ we (i) lower-bound the positive score $s^+$ (the numerator) and (ii) upper-bound each negative score $s_k^-$ (the denominator). We proceed with both bounds.

*(i) Lower-bounding the numerator (positive A–C score).* Assume $\langle \hat{a}_i, \hat{b}_i \rangle \geq 1 - \epsilon$ and $\langle \hat{b}_i, \hat{c}_i \rangle \geq 1 - \epsilon$ from (P). Decompose $\hat{a}_i$ and $\hat{c}_i$ into the part along $\hat{b}_i$ and the residual orthogonal to $\hat{b}_i$:

$$\hat{a}_i = \langle \hat{a}_i, \hat{b}_i \rangle\, \hat{b}_i + \left(\hat{a}_i - \langle \hat{a}_i, \hat{b}_i \rangle\, \hat{b}_i\right),$$
$$\hat{c}_i = \langle \hat{c}_i, \hat{b}_i \rangle\, \hat{b}_i + \left(\hat{c}_i - \langle \hat{c}_i, \hat{b}_i \rangle\, \hat{b}_i\right).$$

By bilinearity, with explicit multiplication and orthogonality indicated,

$$\langle \hat{a}_i, \hat{c}_i \rangle = \left\langle \left[\langle \hat{a}_i, \hat{b}_i \rangle\, \hat{b}_i + \underbrace{\left(\hat{a}_i - \langle \hat{a}_i, \hat{b}_i \rangle\, \hat{b}_i\right)}_{\perp\, \hat{b}_i}\right], \left[\langle \hat{c}_i, \hat{b}_i \rangle\, \hat{b}_i + \underbrace{\left(\hat{c}_i - \langle \hat{c}_i, \hat{b}_i \rangle\, \hat{b}_i\right)}_{\perp\, \hat{b}_i}\right] \right\rangle$$

$$= \langle \hat{a}_i, \hat{b}_i \rangle \langle \hat{c}_i, \hat{b}_i \rangle \underbrace{\langle \hat{b}_i, \hat{b}_i \rangle}_{=1} + \langle \hat{a}_i, \hat{b}_i \rangle \underbrace{\langle \hat{b}_i, \hat{c}_i - \langle \hat{c}_i, \hat{b}_i \rangle\, \hat{b}_i \rangle}_{=0} + \langle \hat{c}_i, \hat{b}_i \rangle \underbrace{\langle \hat{a}_i - \langle \hat{a}_i, \hat{b}_i \rangle\, \hat{b}_i, \hat{b}_i \rangle}_{=0}$$

$$+ \left\langle \hat{a}_i - \langle \hat{a}_i, \hat{b}_i \rangle\, \hat{b}_i, \; \hat{c}_i - \langle \hat{c}_i, \hat{b}_i \rangle\, \hat{b}_i \right\rangle$$

$$= \langle \hat{a}_i, \hat{b}_i \rangle \langle \hat{c}_i, \hat{b}_i \rangle + \left\langle \hat{a}_i - \langle \hat{a}_i, \hat{b}_i \rangle\, \hat{b}_i, \; \hat{c}_i - \langle \hat{c}_i, \hat{b}_i \rangle\, \hat{b}_i \right\rangle.$$

Apply Cauchy–Schwarz to the residual inner product:

$$\langle \hat{a}_i, \hat{c}_i \rangle \geq \langle \hat{a}_i, \hat{b}_i \rangle \langle \hat{c}_i, \hat{b}_i \rangle - \left\| \hat{a}_i - \langle \hat{a}_i, \hat{b}_i \rangle\, \hat{b}_i \right\| \left\| \hat{c}_i - \langle \hat{c}_i, \hat{b}_i \rangle\, \hat{b}_i \right\|.$$

Since $\|\hat{a}_i\| = \|\hat{b}_i\| = \|\hat{c}_i\| = 1$,

$$\left\| \hat{a}_i - \langle \hat{a}_i, \hat{b}_i \rangle\, \hat{b}_i \right\| = \sqrt{1 - \langle \hat{a}_i, \hat{b}_i \rangle^2}, \qquad \left\| \hat{c}_i - \langle \hat{c}_i, \hat{b}_i \rangle\, \hat{b}_i \right\| = \sqrt{1 - \langle \hat{c}_i, \hat{b}_i \rangle^2}.$$

Let $\alpha := \langle \hat{a}_i, \hat{b}_i \rangle$ and $\beta := \langle \hat{c}_i, \hat{b}_i \rangle$; by (P), $\alpha, \beta \in [1 - \epsilon, 1]$. Thus

$$\langle \hat{a}_i, \hat{c}_i \rangle \geq \alpha\beta - \sqrt{1 - \alpha^2}\, \sqrt{1 - \beta^2}.$$

The right-hand side is minimized over $\alpha, \beta \in [1 - \epsilon, 1]$ at $\alpha = \beta = 1 - \epsilon$, giving

$$\langle \hat{a}_i, \hat{c}_i \rangle \geq (1 - \epsilon)^2 - \left(1 - (1 - \epsilon)^2\right) = 2(1 - \epsilon)^2 - 1 = p(\epsilon).$$

*(ii) Upper-bounding the denominator (negative A–C scores).* Let $j \neq i$. Decompose $\hat{a}_i$ and $\hat{c}_j$ along $\hat{b}_i$:

$$\hat{a}_i = \langle \hat{a}_i, \hat{b}_i \rangle \, \hat{b}_i + \left(\hat{a}_i - \langle \hat{a}_i, \hat{b}_i \rangle \, \hat{b}_i\right), \qquad \hat{c}_j = \langle \hat{c}_j, \hat{b}_i \rangle \, \hat{b}_i + \left(\hat{c}_j - \langle \hat{c}_j, \hat{b}_i \rangle \, \hat{b}_i\right).$$

As above, cross terms vanish:

$$\langle \hat{a}_i, \hat{c}_j \rangle = \langle \hat{a}_i, \hat{b}_i \rangle \langle \hat{c}_j, \hat{b}_i \rangle + \langle \hat{a}_i - \langle \hat{a}_i, \hat{b}_i \rangle \, \hat{b}_i, \; \hat{c}_j - \langle \hat{c}_j, \hat{b}_i \rangle \, \hat{b}_i \rangle.$$

Pivot term: $\langle \hat{a}_i, \hat{b}_i \rangle \in [1 - \epsilon, 1]$ and $\langle \hat{c}_j, \hat{b}_i \rangle \leq \eta$, hence

$$\langle \hat{a}_i, \hat{b}_i \rangle \langle \hat{c}_j, \hat{b}_i \rangle \leq \max\{\eta, (1 - \epsilon)\eta\}.$$

Orthogonal term: $\|\hat{a}_i - \langle \hat{a}_i, \hat{b}_i \rangle \, \hat{b}_i\| \leq \sqrt{1 - (1 - \epsilon)^2} = \sqrt{2\epsilon - \epsilon^2}$ and $\|\hat{c}_j - \langle \hat{c}_j, \hat{b}_i \rangle \, \hat{b}_i\| \leq 1$, hence by Cauchy–Schwarz the residual inner product is at most $\sqrt{2\epsilon - \epsilon^2}$. Thus

$$\langle \hat{a}_i, \hat{c}_j \rangle \leq \max\{\eta, (1 - \epsilon)\eta\} + \sqrt{2\epsilon - \epsilon^2} = q(\epsilon, \eta).$$

Combining the numerator and denominator bounds yields the stated inequality for $\ell_i$. Monotonicity follows because $p(\epsilon)$ is strictly decreasing in $\epsilon \in [0, 1)$ while $q(\epsilon, \eta)$ is increasing in each of $\epsilon$ and $\eta$. For the limit: $p(\epsilon) \to 1$ as $\epsilon \to 0$, and $q(\epsilon, \eta) \to -1$ as $(\epsilon, \eta) \to (0, -1)$ since $\max\{\eta, (1 - \epsilon)\eta\} \to -1$ and $\sqrt{2\epsilon - \epsilon^2} \to 0$. Therefore $q(\epsilon, \eta) - p(\epsilon) \to -2$ and $\ell_i \leq \log(1 + Ne^{-2/\tau})$, which tends to 0 only if additionally $\tau \downarrow 0$. $\qquad \square$

## A.1. Discussion of Lemma 3.1

**Implicit assumptions.** The proof treats (P)–(N) as *uniform* worst-case margins over the evaluation set, whereas InfoNCE optimizes an *average*; to instantiate the bounds in practice, one could choose $\epsilon$ and $\eta$ from held-out/evaluation statistics, but this is not the main point of this bound. We assume each anchor $b_i$ has a single intended positive $c_i$; if multiple $c$'s legitimately match $b_i$, exclude those pairs from the negative set or use a multi-positive protocol; otherwise (N) is contradicted. The negative cap $\eta$ must bound all mismatched pairs that appear at test time; if evaluation introduces harder negatives (larger corpus or domain shift), update $\eta$ accordingly. Because cosine $\in [-1, 1]$, the per-sample bound tightens with better margins but does not vanish as $\epsilon \to 0$ and $\eta \to -1$ unless $\tau \to 0$ (cf. Lemma 3.1).

**Requirements for strong transfer** The sufficient condition $p(\epsilon) > q(\epsilon, \eta)$ can fail–or the bound loosen–when (i) supervision is contradictory (true multi-matches treated as negatives), (ii) domain shift makes $\phi_B$ an unstable "ruler", (iii) evaluation negatives are harder than those used to set $\eta$, (iv) residual components orthogonal to $\hat{b}_i$ are *aligned* across $A$ and $C$ (thereby *increasing* $q$), (v) the negative set is very large (the loss scales like $\log(1 + Ne^{(q-p)/\tau})$), or (vi) evaluation uses a different similarity than cosine.

## B. Preprocessing of Evaluation Datasets

We benchmark SPATIALWHISPERER and the baselines on a total of six benchmark datasets.

To strengthen published baselines in our analysis, we initially assessed their performance on two sets of queries. First, the raw cell type labels, and second, a full sentence constructed as *A sample of {label}*. Pathology vision-language models performed stronger on the direct labels, and so we performed evaluations for all models (including ours) with that label set. The comparison of label performance is shown in Table 3.

In the subsections below, we describe any filtering and modifications to ensure semantic coherence of dataset labels, as required for meaningful zero-shot evaluations.

### B.1. PathoCell (13 classes)

PathoCell (Lüscher et al., 2025) comprises 109 colorectal cancer tissue regions from 35 patients, with cell-type labels derived from a CODEX proteomics assay performed in parallel to the H&E imaging (Schürch et al., 2020). The published

*Table 3.* Comparison of Conch and PLIP Terms Performance. **Direct** corresponds to raw labels (cell type names). **Phrase** corresponds to a constructed sentence

| Class Label | Conch Terms | | PLIP Terms | |
|---|---|---|---|---|
| | Direct | Phrase | Direct | Phrase |
| Adipocytes | 0.545 | 0.518 | 0.467 | 0.469 |
| B cells | 0.584 | 0.663 | 0.538 | 0.541 |
| Dendritic cells | 0.525 | 0.557 | 0.485 | 0.478 |
| Granulocytes | 0.631 | 0.431 | 0.428 | 0.413 |
| Macrophages/Monocytes | 0.426 | 0.327 | 0.406 | 0.372 |
| NK cells | 0.437 | 0.337 | 0.395 | 0.345 |
| Nerves | 0.567 | 0.525 | 0.547 | 0.575 |
| Plasma cells | 0.660 | 0.649 | 0.496 | 0.514 |
| Smooth muscle | 0.670 | 0.662 | 0.630 | 0.595 |
| Stroma | 0.515 | 0.494 | 0.363 | 0.338 |
| T cells | 0.426 | 0.459 | 0.502 | 0.476 |
| Tumor cells | 0.497 | 0.613 | 0.506 | 0.505 |
| Vasculature/Lymphatics | 0.611 | 0.595 | 0.579 | 0.510 |
| Mean | 0.546 | 0.525 | 0.488 | 0.472 |

benchmark provides 15 distinct classes, including "other cells" and "background" classes, which we remove. We use the "B cells, T cells, Tumor cells, Stroma, Smooth muscle, Nerves, Plasma cells, Granulocytes, Macrophages/Monocytes, NK cells, Adipocytes, Dendritic cells, and Vasculature/Lymphatics" verbatim as provided in the dataset.

Per-class AUROC across the 109 fields-of-view is computed under the presence-based labeling scheme described in Appendix D.

### B.2. Lizard (3 classes)

Lizard (Graham et al., 2021) provides nucleus-level cell-type labels across 70 colon adenocarcinoma tissue datasets (CRAG, GlaS, DigestPath subsets) with 6 classes: Neutrophil, Epithelial, Lymphocyte, Plasma, Eosinophil, Connective tissue. We patchify and aggregate nucleus labels into per-patch counts, then apply the presence-based AUROC scheme of Appendix D. For the manuscript, we report on 3 reduced classes to capture coarse-grained cell type, complementing the finer-grained PathoCell label set: *Epithelial* (passthrough), *Leukocyte* = $\sum${Neutrophil, Lymphocyte, Eosinophil}, and *Fibroblast* (renamed from Connective tissue). *Plasma* is dropped due to its underrepresentation and scattered nature, which is poorly compatible with the patch-level evaluations in this paper.

### B.3. PanNuke (4 classes)

PanNuke (Gamper et al., 2019) provides nucleus-level labels across 51 datasets spanning 19 organs, with 5 classes: Epithelial, Dead Cells, Connective/Soft tissue cells, Inflammatory, Neoplastic cells. We follow the same patch-aggregation and presence-based AUROC scheme (Appendix D). We drop *Dead Cells* and report the remaining 4 classes.

### B.4. Skin Conditions benchmark

Kriegsmann et al. (2022) curate a 16-class H&E patch benchmark of skin tissue, with a patient-stratified train/validation/test split across 386 patients. We evaluate on the released test split; the train and validation splits are ignored.

The 16 class names are the clinical labels reported by the authors: 12 non-tumor anatomical structures (*Necrosis*, *Skeletal muscle*, *Eccrine glands*, *Vessels*, *Elastosis*, *Chondral tissue*, *Hair follicle*, *Epidermis*, *Nerves*, *Subcutis*, *Dermis*, *Sebaceous glands*) and 4 tumor types (*Squamous cell carcinoma*, *Melanoma*, *Basal cell carcinoma*, *Naevi*).

Unlike the cell-type benchmarks (Appendix D), each Skin Conditions patch carries a single class label, so we report standard macro-averaged one-vs-rest AUROC across the 16 classes (cosine similarity between image and class-name text embeddings).

## B.5. Tabula Sapiens

Tabula Sapiens (The Tabula Sapiens Consortium et al., 2022) was obtained, processed, and used as described in (Schaefer et al., 2025b).

# C. Baselines for cell-type prediction

Here, we describe all published and internal baselines we benchmarked against SPATIALWHISPERER. Table 4 provides the per-class breakdown of the macro AUROC reported in Table 1, across all 7 methods and 20 classes (13 PathoCell + 3 Lizard + 4 PanNuke).

*Table 4.* Per-class AUROC underlying the macro-AUROC summary in Table 1. SPATIALWHISPERER entries are reported as mean $\pm$ std across three training seeds; all other columns are point estimates (deterministic published models, or trained baselines at seed=0). Method details in Appendix C; class definitions in Appendix B.

| Benchmark | Class | SPATIALWHISPERER | Quilt-1M only | UNI2→GF | OmiCLIP$_{short}$ | OmiCLIP$_{ext.}$ | CONCH | PLIP |
|---|---|---|---|---|---|---|---|---|
| PathoCell | B cells | $0.714 \pm 0.011$ | 0.707 | 0.586 | 0.646 | 0.640 | 0.584 | 0.538 |
| PathoCell | Macrophages/Monocytes | $0.687 \pm 0.038$ | 0.403 | 0.740 | 0.471 | 0.390 | 0.426 | 0.406 |
| PathoCell | Adipocytes | $0.498 \pm 0.022$ | 0.526 | 0.500 | 0.511 | 0.518 | 0.545 | 0.467 |
| PathoCell | Dendritic cells | $0.612 \pm 0.006$ | 0.552 | 0.500 | 0.540 | 0.525 | 0.525 | 0.485 |
| PathoCell | T cells | $0.706 \pm 0.025$ | 0.701 | 0.453 | 0.472 | 0.487 | 0.426 | 0.502 |
| PathoCell | Granulocytes | $0.603 \pm 0.030$ | 0.563 | 0.591 | 0.414 | 0.386 | 0.631 | 0.428 |
| PathoCell | NK cells | $0.490 \pm 0.029$ | 0.451 | 0.362 | 0.450 | 0.463 | 0.437 | 0.395 |
| PathoCell | Nerves | $0.537 \pm 0.009$ | 0.591 | 0.468 | 0.515 | 0.502 | 0.567 | 0.547 |
| PathoCell | Plasma cells | $0.637 \pm 0.010$ | 0.596 | 0.585 | 0.490 | 0.469 | 0.660 | 0.496 |
| PathoCell | Smooth muscle | $0.717 \pm 0.016$ | 0.637 | 0.691 | 0.504 | 0.508 | 0.670 | 0.630 |
| PathoCell | Stroma | $0.629 \pm 0.011$ | 0.599 | 0.681 | 0.445 | 0.507 | 0.515 | 0.363 |
| PathoCell | Tumor cells | $0.531 \pm 0.039$ | 0.535 | 0.500 | 0.410 | 0.343 | 0.497 | 0.506 |
| PathoCell | Vasculature/Lymphatics | $0.718 \pm 0.004$ | 0.499 | 0.500 | 0.514 | 0.475 | 0.611 | 0.579 |
| Lizard | Epithelial | $0.906 \pm 0.027$ | 0.757 | 0.805 | 0.631 | 0.617 | 0.562 | 0.615 |
| Lizard | Leukocyte | $0.586 \pm 0.041$ | 0.706 | 0.450 | 0.506 | 0.542 | 0.613 | 0.484 |
| Lizard | Fibroblast | $0.756 \pm 0.020$ | 0.462 | 0.730 | 0.694 | 0.781 | 0.669 | 0.484 |
| PanNuke | Epithelial | $0.528 \pm 0.010$ | 0.708 | 0.629 | 0.465 | 0.438 | 0.733 | 0.538 |
| PanNuke | Connective/Soft tissue cells | $0.696 \pm 0.010$ | 0.723 | 0.696 | 0.535 | 0.525 | 0.650 | 0.639 |
| PanNuke | Inflammatory | $0.714 \pm 0.021$ | 0.654 | 0.570 | 0.508 | 0.508 | 0.609 | 0.577 |
| PanNuke | Neoplastic cells | $0.811 \pm 0.016$ | 0.766 | 0.500 | 0.573 | 0.632 | 0.422 | 0.581 |

## C.1. Pathology vision–language models (CONCH, PLIP)

CONCH (Lu et al., 2024) and PLIP (Huang et al., 2023) are CLIP-style image–text contrastive foundation models for histopathology. CONCH was trained on 1.17M H&E patch–caption pairs curated from PubMed Central open-access articles using a CoCa-style contrastive-plus-captioning objective; PLIP was trained on 208K patch–caption pairs from pathology Twitter (OpenPath corpus) on a CLIP ViT-B/32 backbone. Like SPATIALWHISPERER, these methods score a class by cosine similarity between an H&E patch embedding and the text embedding of a class prompt; the un-thresholded similarities serve as AUROC inputs. Both are deterministic single-pass evaluations.

## C.2. Trained bimodal $\mathcal{I} \leftrightarrow \mathcal{T}$ baseline (Quilt-1M only)

To isolate performance effects from our encoder modules from the proposed transitive training mechanism, we include a bimodal $\mathcal{I} \leftrightarrow \mathcal{T}$ baseline trained on natural-language pathology descriptions in QUILT-1M (Ikezogwo et al., 2023). This baseline operates analogously to PLIP/CONCH, while using the same single-stage contrastive architecture and training protocol as SPATIALWHISPERER.

## C.3. UNI2→Geneformer two-stage pipeline

As an intuitive baseline to the problem of cell-level H&E annotation, we trained a two-stage pipeline (image → predicted expression → cell type) on the same paired image–expression data as our presented model, while reusing the same image and expression encoders.

The first stage maps H&E patches to predicted gene expression. A frozen UNI2 encoder (Chen et al., 2024) produces

a 1536-dimensional patch embedding, which is decoded by a 2-layer MLP with residual connections and LayerNorm (width 2048) to predict expression of 17,851 genes (intersection of HEST-1K and Geneformer's vocabulary). The decoder was trained on ∼900K Visium spot-level image–expression pairs from HEST-1K (Jaume et al., 2024) under MSE on $\log(\text{counts} + 1)$ for 4 epochs (lr=$10^{-3}$, batch=256, AdamW with cosine schedule). On HEST-bench held-out evaluation (10 organs × ∼50 curated panel genes), the decoder reaches a cross-organ median per-gene Pearson correlation of $0.47$ (organ range 0.31–0.71).

The second stage classifies cell types from predicted expression. Geneformer (Theodoris et al., 2023) (12L-30M, frozen backbone) with a learnable linear head was fine-tuned on labeled CELLxGENE Census transcriptome profiles (CZI Single-Cell Biology Program et al., 2023) (482 classes, 8 epochs, lr=$10^{-4}$) as described in (Schaefer et al., 2025b). At inference time, the classifier output probabilities over the 482 training classes are mapped to each benchmark's class space via a regex-word-boundary mapping (e.g. `plasma cell` → "Plasma cells" in PathoCell; `T cell, B cell, NK cell, . . .` → "Lymphocyte" in Lizard). Probabilities for multiple training classes mapping to the same benchmark class are summed.

### C.4. OmiCLIP / Loki Annotate

OmiCLIP (Chen et al., 2025a) is a contrastive image–gene-expression foundation model trained on ∼2.2M paired H&E patches and "gene-expression sentences" (space-separated lists of top-expressed gene symbols per Visium spot). The Loki Annotate pipeline performs zero-shot cell-type annotation by representing each cell type as its marker-gene set and scoring cosine similarity between an H&E patch embedding (CoCa ViT-L/14 image encoder) and the marker-gene-set embedding (expression sentence encoder). By contrast, our trimodal framework jointly aligns image, gene expression, and textual *annotations* such as cell types. We followed Loki Annotate's marker-gene variant with two marker-list densities per benchmark:

- **Short marker list.** 8–12 canonical markers per class (e.g. `CD79A MS4A1 CD19 BANK1 IGHM IGHD CD22 BLK` for B cells), curated from PanglaoDB, the Human Protein Atlas, and standard colorectal cancer scRNA-seq marker panels. This matches the gene-set density of the Loki Annotate tutorial.

- **Extended marker list.** 25–30 markers per class, ordered by marker specificity, designed to fit OmiCLIP's training distribution (Visium pseudobulk gene sentences, ∼76-token cap per class).

For the colon-context benchmarks (PathoCell and Lizard, the latter sourced largely from colon adenocarcinoma datasets such as CRAG, GlaS, and DigestPath), we used a pan-epithelial backbone (EPCAM, CDH1) extended with colon-specific differentiation markers (KRT20, MUC2, CDX2, VIL1, CEACAM5) for the epithelial-lineage class in each benchmark—PathoCell's "Tumor cells" and Lizard's "Epithelial". The two panels overlap heavily because tissue-of-origin markers persist in adenocarcinoma. We did not separately curate a tumor-specific panel (proliferation or oncogene readouts) since each benchmark treats the class as a single bucket. For PanNuke's "Epithelial" class, which spans samples from many organs, we replaced the colon-specific markers (CDX2, MUC2, VIL1, CEACAM5) with a pan-epithelial backbone (EPCAM, CDH1) extended by a diversified keratin set: KRT5/14 for stratified squamous epithelium (skin, esophagus, cervix), KRT7/17 for glandular and ductal epithelia (lung, breast, biliary), and KRT8/18/19 for simple/columnar epithelia (gut, liver, kidney).

Implementing this baseline highlighted two challenges for marker-gene proxies. First, the panels must be hand-designed (not learned from data), so the curator's choice of markers propagates directly into the score. Second, even a well-curated panel is interpreted by an encoder that was pretrained on Visium-spot gene-expression sentences rather than on cell-type marker-set queries, so there is no guarantee that the model weighs the supplied genes the way the curator intended. Consequently, our efforts to design different panels to optimize this baseline led to underwhelming results. In contrast, SPATIALWHISPERER optimizes the relation between semantic labels, such as cell types, and the H&E image representation.

## D. Discussion of metrics and comparison

The patch-level resolution of our model implies a soft-labeled prediction scenario where most patches contain cells of various types (i.e., classes). An intuitive metric to capture this would be the Kullback–Leibler divergence over the categorical distribution of cell types. Indeed, this direct metric exhibits strongly improved performance of SPATIALWHISPERER compared to published baselines (Fig. 8). But it is challenging to interpret.

To measure class-level (i.e., cell type) performance, we found AUROC to be most expressive, as it considers the model's

continuous CLIP scores in light of the patch-level cell-type composition. We use a *presence-based* (multi-positive) labeling, where for each class $j$, a patch is treated as positive if it contains *any* cells of class $j$. Per-class AUROC is computed one-vs-all by ranking patches by their predicted score for class $j$. AUROC is then macro-averaged across datasets within a class, and then across classes to a single per-benchmark scalar. This same presence-based formulation is used for all per-class evaluations in the manuscript (PathoCell, Lizard, and PanNuke). We also evaluated the alternative "most-represented class" (argmax-onehot) setting, in which each patch is labeled by its single most abundant cell type, and observed similar overall performance.

Other scores, such as the commonly employed $F_1$, are less suited for the categorical label setting. $F_1$ rewards models' predictions *only* in the case where the model assigns the highest score to the model with the highest abundance, neglecting any other reasonable outputs as equally wrong. In line with this, we observed less coherent and generally weaker results for $F_1$ scores across all models (see Table 5).

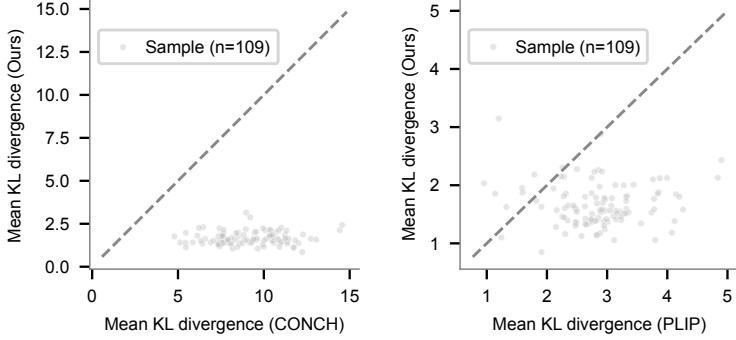

*Figure 8.* Kullback–Leibler divergence between patch-level predicted and ground truth label distribution. Shown are means across patches for each of 109 sample images in the PathoCell benchmark (Lüscher et al., 2025; Schürch et al., 2020).

*Table 5.* Zero-shot cell type prediction performances (F1 score). Best per row is **bold**; second-best is underlined. SPATIALWHISPERER model trained with $seed = 0$ was assessed.

| Cell Type | SPATIALWHISPERER | CONCH | PLIP |
|---|---|---|---|
| Adipocytes | 0.0289 | 0.0000 | **0.0685** |
| B cells | **0.0690** | 0.0000 | 0.0427 |
| Dendritic cells | **0.0051** | 0.0000 | 0.0000 |
| Granulocytes | 0.0363 | **0.0898** | 0.0000 |
| Macrophages/ Monocytes | 0.0653 | **0.1319** | 0.0000 |
| NK cells | 0.0000 | 0.0000 | 0.0000 |
| Nerves | 0.0000 | **0.0053** | 0.0016 |
| Plasma cells | 0.0599 | **0.1086** | 0.0000 |
| Smooth muscle | **0.2966** | 0.2058 | 0.0200 |
| Stroma | **0.0535** | 0.0000 | 0.0245 |
| T cells | 0.0169 | 0.0000 | **0.0580** |
| Tumor cells | 0.0982 | **0.1074** | 0.0010 |
| Vasculature/ Lymphatics | **0.1009** | 0.0000 | 0.0174 |
| **Mean** | **0.0639** | 0.0499 | 0.0180 |

## E. Effects of text annotation harmonization

The textual annotations associated with images (Ikezogwo et al., 2023) and with gene expression profiles (Schaefer et al., 2025b) are produced independently and differ in style and biological focus, weakening the shared-modality overlap that

transitive alignment relies on (Section 3.3, "Effect of imperfect overlap in $\mathcal{G}$"). To partially compensate this divergence, we harmonized the QUILT-1M captions so their style and vocabulary match those of the $\mathcal{G} \leftrightarrow \mathcal{T}$ corpus. We quantified the effect of this along two axes: (i) image–text retrieval under a held-out trimodal model (already discussed in Section 4, "Transitive Representation Learning Requires Overlapping Modalities"), and (ii) cell-type prediction under a newly trained trimodal model that ingests all three paired datasets, with QUILT-1M provided in its harmonized form (described below).

### E.1. Reformatting prompt and processing

We used GPT OSS 120B (OpenAI et al., 2025) instructing the model to rewrite "histopathological image descriptions into biological sample descriptions" with the goal to "transform descriptions that focus on histological features and pathological findings into descriptions that emphasize the biological sample, cell types, and experimental context", drawing from a list of 20 *bona fide* examples sampled from the transcriptome–text dataset (Schaefer et al., 2025b). The model was supplied the original QUILT-1M label together with more extensive textual contexts produced by the original annotation pipeline (Ikezogwo et al., 2023). The prompt is in the box below.

---

**QUILT-1M caption harmonization prompt (verbatim)**

You are a biomedical scientist with broad biomedical experience in histopathology and single-cell omics.
You are tasked with rewriting histopathological image descriptions into biological sample descriptions. Your goal is to transform descriptions that focus on histological features and pathological findings into descriptions that emphasize the biological sample, cell types, and experimental context.
**INSTRUCTIONS:**

1. Identify the underlying biological sample, cell type, or tissue from the histological description

2. Infer the most likely experimental context, cell line, tissue origin, or clinical sample type

3. Include relevant biological characteristics (age, gender, treatments, conditions) when they can be reasonably inferred

4. Maintain all biomedical information (including disease) while shifting focus from pathological observation to sample description

5. If multiple cell types are mentioned, focus on the primary biological sample being studied

6. If some of the provided information (e.g. in the provided full-text seems nonsensical then leave it out

7. Focus on using the `caption` and use the other information only in an auxiliary manner.

8. Think step-by-step about the biological sample identification and context inference, then provide the rewritten description as your final answer.

**INPUT STYLE AND INFORMATION (what you'll receive):**

- JSON with multiple fields (*caption*, full-text, pathology information, region-of-interest text).

- Focuses on histological patterns, pathological findings, and diagnostic observations

- Examples: "Red blood cells that appear fragmented", "Identification of yeast with capsule", "MPGN pattern with occluded glomerular lumen"

**OUTPUT STYLE (what you should produce):**

- Focuses on the biological sample, cell type, tissue origin, and experimental context.

- Produce a single unformatted prose string output caption.

- Below are some examples of output captions in the style you should produce. Especially copy their focus on a "sample" or a set of "cells".

1. MDA-MB-231 cell line derived from a breast cancer tumor, treated with TRIM11-sgRNA1 and KDM5C-sgRNA2 to knockdown TRIM11 and KDM5C.

2. T-cell acute lymphoblastic leukemia (T-ALL) cells isolated from NSG spleen.

3. A sample of peripheral blood from a coronary arterial calcification (CAC) patient.

---

4. 50-year-old male prostate cancer cells (LNCaP) treated with Ponatinib, obtained from ATCC.

5. 56-week-old male PBMC cells from a site with low malaria transmission, taken from a subject who did not have malaria before day 365 post-vaccination and received a comparator vaccine during the M0 visit. The cells were stimulated with csp. The subject's Waz and Haz scores were $-3.22$ and $-2.42$, respectively, and their hemoglobin level was 7.7 g/dL.

6. Human induced pluripotent-stem-cell derived embryoid body treated with Hydroxyurea at a concentration of $2.50 \times 10^{-5}$ for factor analysis and model building as part of a study on teratogenicity.

7. 70.3-year-old meningioma tissue sample from UCSF with a grade of 2.

8. 26Sp1d_B06 are stem cells cultured for 26 days, with SOX2+ cre line.

9. 60-year-old male Meningioma patient (M2) with Intracranial Meningioma (cancerous growth in the meninges, the membranes that cover the brain and spinal cord).

10. 0.1 ppm 1-Bromopropane treated HBEC cell line.

11. 48-hour treated cardiac myocytes derived from induced pluripotent stem cell line MSN02, with a concentration of 3 uM Carfilzomib, in skin tissue, with RIN value of 9.4.

12. The sample is a cell line of glioblastoma cancer stem cells, specifically the GliNS2 cell line, with a grade IV designation and aneuploid characteristics. It is a cell line derived from a glioblastoma, a type of aggressive brain cancer, and has been cultured for study. No specific treatments have been applied to this cell line.

13. 48-hour cultured primary peripheral blood mononuclear cells (PBMCs) from a male donor, assayed using Smart-seq2 technology.

14. HT-29 colon adenocarcinoma cell line derived from a Caucasian adult female, genotype Trim28 $-/-$, treated with vehicle.

15. 15-day old, wild-type induced pluripotent stem cell-derived cardiomyocytes, grown in vitro.

16. Primary cells from a human peripheral blood sample, specifically memory B cells, that were derived from PBMCs (peripheral blood mononuclear cells) of a donor with spontaneous clearance of HCV (hepatitis C virus) infection.

17. BJ Fibroblasts, 8 days post induced viral expression of OSKM treatment.

18. Sample MM38_53 is from a female patient with multiple myeloma, derived from bone marrow cells that have been positively selected for CD138.

19. HepG2 cells exposed to 1.2 ug/mL citrate treatment, a type of hepatocellular carcinoma cell line derived from a Caucasian male liver.

20. B-Lymphocyte cell line, GM18915, from Yoruba ethnicity with wildtype genotype, untreated.

## E.2. Effect on cell-type prediction

To test the effect of harmonization on cell-type prediction, we trained two new versions of SPATIALWHISPERER, once with the harmonized QUILT-1M captions included, and once with the original ones, while holding all other training data, hyperparameters, and the seed (=0) fixed. We then evaluated zero-shot patch-level cell-type prediction on PathoCell.

Adding the original QUILT-1M $\mathcal{I} \leftrightarrow \mathcal{T}$ data to the transitive $\mathcal{T} \leftrightarrow \mathcal{G} + \mathcal{G} \leftrightarrow \mathcal{I}$ data degrades mean AUROC from 0.630 to 0.609 ($-3.3\%$; Table 6, seed=0). Replacing the raw captions with the harmonized version reverses this: the trimodal model recovers to 0.645, slightly exceeding SPATIALWHISPERER. This empirically validates the prediction in Section 3.3 ("Implications for training on all three edges"): when the bridging text modality is harmonized to a consistent style across the two paired datasets, all three modality edges contribute usefully.

## E.3. Effect on tissue-of-origin prediction

For comparison, we ran the same three trimodal-ablation models on a complementary 19-class tissue-of-origin task on PanNuke (sample-level macro AUROC across 2,401 samples; seed=0). The bimodal $\mathcal{T} \leftrightarrow \mathcal{G} + \mathcal{G} \leftrightarrow \mathcal{I}$ bridge attains 0.586; adding original QUILT-1M $\mathcal{I} \leftrightarrow \mathcal{T}$ substantially raises macro AUROC to 0.646, and harmonization slightly further to 0.649.

*Table 6.* Effect of QUILT-1M text harmonization on zero-shot cell-type prediction (PathoCell, 13 cell types, $n = 109$ samples, mean presence-based AUROC; seed=0). Best per row is **bold**; second-best is underlined. "+QUILT-1M (raw)" and "+QUILT-1M (harmonized)" denote training-data extensions of SPATIALWHISPERER that additionally ingest the QUILT-1M $\mathcal{I} \leftrightarrow \mathcal{T}$ pairs in raw and harmonized form, respectively. All models were trained with seed=0.

| Cell Type | SPATIALWHISPERER | +QUILT-1M (raw) | +QUILT-1M (harmonized) |
|---|---|---|---|
| Adipocytes | 0.473 | 0.491 | **0.508** |
| B cells | **0.726** | 0.724 | 0.678 |
| Dendritic cells | 0.611 | 0.604 | **0.622** |
| Granulocytes | 0.636 | 0.538 | **0.650** |
| Macrophages/Monocytes | 0.712 | 0.731 | **0.774** |
| NK cells | 0.457 | 0.430 | **0.524** |
| Nerves | 0.541 | **0.609** | 0.560 |
| Plasma cells | 0.639 | **0.666** | 0.640 |
| Smooth muscle | 0.734 | 0.710 | **0.761** |
| Stroma | **0.633** | 0.546 | 0.631 |
| T cells | 0.734 | 0.705 | **0.752** |
| Tumor cells | 0.575 | 0.549 | **0.613** |
| Vasculature/Lymphatics | **0.720** | 0.618 | 0.679 |
| Mean | 0.630 | 0.609 | **0.645** |

Combined with the cell-type result above, the harmonized trimodal model is the only configuration of the three that improves over the bimodal bridge on both fine-grained cellular and coarse tissue-of-origin prediction.

## F. Sensitivity to training hyperparameters

### F.1. Encoder choice and freezing configuration

To understand the performance of our method subject to established hyperparameters, we evaluated distinct freezing and encoder parameters.

**Freezing axis** Following the LiT (Zhai et al., 2022) convention, a three-letter code for encoders $(\phi_{\mathcal{G}}, \phi_{\mathcal{T}}, \phi_{\mathcal{I}})$ specifies locked ($L$) versus unfrozen ($U$) towers. We evaluate $LUL$ (SPATIALWHISPERER), $LLL$ (only projection heads trained), and $LLU$ (image tower unfrozen). We attempted $ULL$ (transcriptome unfrozen), but the unlocked Geneformer implementation occupied more than 80GB of RAM at our experimental batch sizes.

**Encoder axis** We swap the bridge encoder Geneformer (Theodoris et al., 2023) for an 8-layer UCE (Rosen et al., 2023).[2] Image and text encoders are held at UNI2 and BioBERT.

**Results** Table 7 reports macro AUROC on PathoCellBench (CRC; 13 classes; presence-based aggregation of Appendix D). Training only the projection heads ($LLL$) costs 8.3 points absolute, confirming that text-encoder optimization is the dominant source of gain on this task, consistent with prior reports in CellWhisperer (Schaefer et al., 2025b) and LiT (Zhai et al., 2022). Unfreezing the image tower ($LLU$) costs 5.4 points absolute relative to $LUL$.[3] The Geneformer $\rightarrow$ UCE swap yields comparable performance (1.3 points absolute below $LUL$), aligned with previous observations (Schaefer et al., 2025b).

### F.2. Composite-loss weights

The training objective sums three modality-pair InfoNCE terms with scalar weights $(\lambda_{\mathcal{I} \leftrightarrow \mathcal{T}}, \lambda_{\mathcal{I} \leftrightarrow \mathcal{G}}, \lambda_{\mathcal{T} \leftrightarrow \mathcal{G}})$. While SPATIALWHISPERER is trained with fixed $1.0$ values, we test the sensitivity of downstream performance to this parameter.

---

[2]Checkpoint https://huggingface.co/KuanP/uce-cxg-2025-baseline-8l-512d

[3]$LLU$ used per-step batch 64 with gradient accumulation 8 (effective batch 512) to fit on an 80 GB H100; all other configurations use per-step batch 512. The contrastive InfoNCE loss therefore sees fewer negatives per step in $LLU$, which can independently affect performance.

*Table 7.* Hyperparameter ablation: effect of freezing configuration and bridge-encoder choice on cell-type prediction (PathoCellBench CRC; 13 classes). All models are trained with seed=0 for one full epoch. *Macro AUROC* averages per-class presence-based AUROC across the 13 classes. *Macro F1* and *Recall@5* are averaged across the 109 fields-of-view. Best per column **bold**.

| Configuration | Macro AUROC | Macro F1 | Recall@5 |
|---|---|---|---|
| *LUL* Geneformer (like SPATIALWHISPERER) | **0.6259** | **0.1061** | 0.5575 |
| *LLL* (proj. heads only) | 0.5428 | 0.0878 | 0.5350 |
| *LLU* (image unfrozen) | 0.5721 | 0.0967 | **0.5788** |
| *LUL* UCE bridge | 0.6134 | 0.0615 | 0.5114 |

**Design**   We sweep the ratio $\lambda_{\mathcal{T}\leftrightarrow\mathcal{G}} : \lambda_{\mathcal{I}\leftrightarrow\mathcal{G}}$ over five points (4:1, 2:1, 1:1, 1:2, 1:4) on the $\mathcal{T}\leftrightarrow\mathcal{G}+\mathcal{G}\leftrightarrow\mathcal{I}$ training mix used in the main paper (cellxgene_census, archs4_geo, hest1k; Quilt-1M excluded, so $\mathcal{L}_{\mathcal{I}\leftrightarrow\mathcal{T}}$ is dormant). Each configuration trains for one full epoch on identical data with identical optimizer state; only the per-pair coefficients in $\mathcal{L}$ differ. The lambdas are applied directly inside the contrastive loss as a weighted mean over active pairs, so the gradient magnitude is invariant to the overall scale of $\boldsymbol{\lambda}$.

**Results**   Table 8 reports macro AUROC on PathoCell. The spread across the full $16\times$ ratio range is 0.0042 absolute. Performance varies monotonically and weakly with the ratio, with a mild penalty for down-weighting the text–gene pair (1:4). The equal-weight setting used SPATIALWHISPERER sits within 0.001 of the best point.

*Table 8.* Sensitivity to composite-loss weights on PathoCellBench (CRC; 13 classes; seed=0; one full epoch; same data and step count across rows). Macro AUROC follows the same presence-based aggregation as Appendix D. Best per column **bold**.

| $\lambda_{\mathcal{T}\leftrightarrow\mathcal{G}} : \lambda_{\mathcal{I}\leftrightarrow\mathcal{G}}$ | Macro AUROC | Macro F1 | Recall@5 |
|---|---|---|---|
| 4:1 | 0.6240 | 0.1064 | **0.5750** |
| 2:1 | **0.6241** | 0.1036 | 0.5652 |
| 1:1 (main paper) | 0.6234 | 0.1034 | 0.5621 |
| 1:2 | 0.6222 | 0.1022 | 0.5650 |
| 1:4 | 0.6199 | 0.1053 | 0.5671 |

