# OpenReview forum: "Transitive Representation Learning Enhances Histopathology Annotation"
_ICML.cc/2026/Conference — ICML 2026 regular_

### Official Review · Reviewer_ZBSb · 2026-03-11

**Soundness:** 2
**Presentation:** 3
**Significance:** 3
**Originality:** 2
**Overall Recommendation:** 4
**Confidence:** 3

**Summary:**

The paper investigates contrastive learning in a transitive setting, i.e. where a mapping A<->C can be learned by training on datasets with paired mappings A<->B and B<->C. The setup is tested in a medical scenario, for learning a mapping between histopathology patches and text, from paired data of histopathology/spatial transcriptomics and text/spatial transcriptomics. That is, spatial transcriptomics provides a bridge between histopathology and text, such that a mapping can be learned without paired histopathology/text data. An analysis for motivating the transitive learning setting is provided, considering aspects of the transitive learning and overlap between datasets for training. Experiments are provided to demonstrate that: 1) zero-shot cell type prediction can be improved using the transitive learning, as compared to using existing histopathology-text models, 2) trimodal training can be used to improve over bimodal training in low-data scenarios, and 3) alignment of text prompts improves similarity by increasing the overlap between modalities.

**Compliance With Llm Reviewing Policy:**

Affirmed.

**Final Justification:**

Additional experiments provided in the rebuttal address my three main comments regarding i) impact of harmonization for trimodal learning, ii) task specific impact including tissue-level prediction, and iii) impact of prompt-design for zero-shot classification. I think that including these results, as well as discussing prior work on transitive learning, would strengthen the paper and make it suitable for acceptance.

**Key Questions For Authors:**

1. There are some arguments for why trimodal learning did not improve the results over the transitive setting for the first experiment. Then, in the third experiment, text transformation is demonstrated to increase the overlap between datasets. To me, a natural experiment to perform would be to see if this transformation improves on the results in the first experiment, by using trimodal learning with the transformed text prompts. Why wasn't this tested? Would it be too computationally expensive?

2. Another aspect of the lack of improvement in the first experiment with trimodal training is the task itself. Here, cell type classification is considered, such that the histopathology/text dataset is not likely to improve due to not being on cell level. Thus, it would be valuable to test for other tasks, such as tissue level classification. Would the transitive learning still be successful and would trimodal learning improve performance?

3. For the first experiment, the text prompts used for zero-shot classification are the raw class labels (Appendix B shows that formulating a simple phrase did not improve the results). However, is it possible to formulate more sophisticated prompts that could improve the results? For example, depending on the training data used for the compared methods, different prompt designs could be optimal, and a similar alignment as in the text harmonization experiments (Section 4.3) could be considered.

4. While the manuscript in general is well-written and organized, the presentation of the data setup is somewhat confusing to me. The datasets used for the first experiment are presented early on, in table 1. Then, for the second experiment, the datasets are presented together with the results, in table 3. Why is this? It would be easier to read and understand if the datasets were presented in the same place.

**Limitations:**

* The discussion around limitations brings up many valuable considerations. Especially, the task considered is well-suited for transitive learning by focusing on cell-level separation, and it is unclear how it will generalize to other tasks. For example, is it possible to learn the image-text task in Fig. 6 using only data for image-gene expression and gene expression-text?

* Are there any other types of biases that might be induced in transitive learning as compared to direct bimodal learning?

**Strengths And Weaknesses:**

Strengths:
+ Interesting investigation of the transitive learning setting, which should be a useful strategy in data-limited scenarios within medical applications.

+ Experiments show promising results, especially for zero-shot cell type classification of histopathology patches where the transitive setting provides better granularity than a direct mapping from histopathology to text.

Weaknesses:
- The paper gives the impression that the proposed setting has not been explored previously, which does not appear to be true. For example, there is this previous paper which was not referenced:
	Tran, M., Dicente Cid, Y., Lahiani, A., Theis, F., Peng, T., & Klaiman, E. (2023). Training transitive and commutative multimodal transformers with loretta. Advances in Neural Information Processing Systems, 36, 42918-42931.

- The methodological contributions are limited, which means that the contribution relies to a large extent on results.

- While the experiments are promising, they are also somewhat limited. It is unclear what is practically required for successful learning, and how it generalizes to different tasks/models/data. For example, it would be valuable with some form of ablation study, e.g., exploring the impact of: encoder (pre-training and architecture of the added MLP layers), freezing/fine-tuning encoders (two are freezed, while one is fine-tuned), strength of different loss terms (Eq. 1), data filtering, etc.

---

> ### Author Rebuttal · Authors · 2026-03-31
>
> We thank the reviewer for the thoughtful review and for recognizing the promise of transitive learning for histopathology. In response to their concerns, we added new analyses on harmonized trimodal training, tissue-level prediction, prompt robustness, and the practical conditions under which trimodal learning helps or hurts.
>
> **Re W1/W2: "The paper gives the impression that the proposed setting has not been explored previously" and "the methodological contributions are limited."**
>
> We thank the reviewer for making us aware of Tran et al. (NeurIPS 2023), which we will cite together with already mentioned prior work (Girdhar et al., 2023, CVPR).
>
> That method demonstrates a two-step alignment through intermediary token-reproduction operating on weak supervision (class-level pairing). While this strategy is not directly applicable to our setting, **we now compare to a conceptually stronger 2-stage H\&E→transcriptome→cell-type prediction baseline** that leverages the datapoint-level pairing in our data (AUROC 0.550 vs **0.630 (ours)**).
>
> In addition, we compare against two more benchmarks:
>
> |  | PLIP | CONCH | Ours |
> | :---- | :---- | :---- | :---- |
> | PanNuke | 0.588 | 0.624 | **0.689** |
> | Lizard | 0.528 | 0.581 | **0.764** |
>
> **Re Q1: Why wasn't trimodal learning with the transformed text prompts tested?**
>
> Great point. We have now run this experiment (see W3 response), confirming that harmonization improves alignment.
>
> **Re Q2|L1: Would transitive learning improve performance for other tasks like tissue classification and those shown in Fig. 6?**
>
> Indeed, Fig. 6 already shows purely transitive learning (red line at x=0.0). For these tasks, we observed that the transitive advantage was fully exhausted by sufficient target training data (see x=1.0). We will clarify this in the main text.
>
> We now also performed a *19-tissue zero-shot prediction* benchmark on PanNuke:
>
> | Model (training data) | Tissue prediction (PanNuke; AUROC) |
> | :---- | :---- |
> | Ours: T↔G+G↔I | 0.586 |
> | T↔G+G↔I+I↔T | 0.646 |
>
> *Interpretation:* Transitive learning helps some tasks more than others. Training on all three paired datasets integrates “the best of both worlds” (e.g. high tissue-level *and* cell-level performance (see W3)).
>
> **Re W3: "It is unclear what is practically required for successful learning"**
>
> To provide better guidance, we will include the finding above (Q2|L1) and three further new findings in the revised manuscript:
>
> 1. New models trained on all three paired modalities confirm that **modality harmonization impacts performance**.
>
> | Model (training data) | Cell type prediction (PathoCell; AUROC) |
> | :---- | :---- |
> | Ours: T↔G+G↔I | 0.630 |
> | T↔G+G↔I+I↔T | 0.609 |
> | T↔G+G↔I+I↔T (harmonized) | **0.645** |
>
> 2. In a new ablation, we varied the training signal from T↔G and G↔I datasets, indicating that balanced training (λ=1) is a reasonable default.
>
> | Config | Macro AUROC |
> | :---- | :---- |
> | Balanced (1:1) | 0.624 |
> | Image-heavy (1:2) | 0.620 |
> | Text-heavy (2:1) | 0.605 |
>
> 3. Additional encoder/freezing ablations will follow.
>
> **Re L2: "Are there any other types of biases that might be induced in transitive learning as compared to direct bimodal learning?"**
>
> We highly appreciate this comment and will refine our Limitations with two points:
>
> 1. Susceptibility to style/formatting of modality overlap (see W3 response)
> 2. There is a bottleneck in the bridge modality that restricts information transfer, which is most relevant for weakly-expressive modalities.
>
> **Re Q3: "Is it possible to formulate more sophisticated prompts that could improve the results?"**
>
> We tested this directly:
>
> | Format | AUROC |
> | :---- | :---- |
> | ***"a tissue sample containing {label}"*** | **0.647** |
> | "gene expression profiling reveals {label}" | 0.633 |
> | "H\&E tissue image showing {label}" | 0.633 |
> | *"{label}"* | 0.630 |
> | "histopathology image of {label}" | 0.628 |
>
> *Interpretation:* Prompt engineering may help slightly but does not change the qualitative conclusion.
>
> **Re Q4: "The presentation of the data setup is somewhat confusing to me”**
>
> We agree this is harder to follow than necessary. In the revision, we will consolidate all training datasets and evaluation benchmarks into one experimental-setup block before the results, separating (i) training datasets for the three modality pairs from (ii) benchmark datasets.
>
> Overall, these additions clarify the paper's novelty relative to prior transitive-learning work and provide the practical characterization this reviewer requested. Our new analyses establish three **actionable insights for practitioners informing about practical data curation considerations**: (1) transitive learning requires task-relevant bridge semantics, whereas misaligned auxiliary supervision hurts, (2) shared-modality harmonization is a controllable lever that improves both bridge-mediated retrieval and downstream prediction, and (3) the method is largely robust to imbalanced paired modality training signal.

---

> > ### Author Rebuttal · Reviewer_ZBSb · 2026-04-02
> >
> > Thanks for the clarifications and additional experiments! These address my three main comments regarding i) impact of harmonization for trimodal learning, ii) task specific impact including tissue-level prediction, and iii) impact of prompt-design for zero-shot classification. I think that including these results, as well as discussing prior work on transitive learning, would strengthen the paper, and I will adjust my recommendation accordingly.

---

> > > ### Author Response · Authors · 2026-04-03
> > >
> > > We appreciate the time and thoughtful review.

---

### Official Review · Reviewer_rVc4 · 2026-03-12

**Soundness:** 3
**Presentation:** 3
**Significance:** 3
**Originality:** 3
**Overall Recommendation:** 4
**Confidence:** 4

**Summary:**

This paper proposes an innovative trimodal contrastive learning framework that leverages spatial gene expression profiles as a bridge modality to enable transitive representation learning between histopathology images and natural language descriptions, effectively addressing the scarcity of fine-grained cellular annotations in computational pathology. The approach is supported by a sound theoretical derivation of an unobserved InfoNCE loss bound and demonstrates strong clinical utility in zero-shot cell type prediction and low-data retrieval tasks. To further enhance methodological rigor and completeness, the study would benefit from incorporating strictly controlled internal bimodal baselines, quantitative spatial evaluation metrics, and comprehensive hyperparameter ablation studies.

**Compliance With Llm Reviewing Policy:**

Affirmed.

**Final Justification:**

The authors' rebuttal addresses my main concerns by providing crucial new experiments, including controlled baselines and a loss weight ablation. These additions significantly strengthen the paper's empirical claims, and I will maintain my acceptance score.

**Key Questions For Authors:**

1. To properly isolate the contribution of the proposed trimodal architecture from the benefits of using large-scale external gene expression datasets, could you provide zero-shot classification results for a strictly controlled internal bimodal baseline? Specifically, how does an Image-Text model trained exclusively on the same data splits (e.g., QUILT-1M) compare to your method? Additionally, how does a "two-hop" pipeline baseline (Image $\rightarrow$ nearest Gene via retrieval $\rightarrow$ Text scoring) perform using identical backbones?


2.  Lemma 3.1 relies on strict assumptions such as uniform margins and single positive pairs. Given that real-world histopathology data frequently features multi-positives (co-existing cell types in a patch) and label ambiguity, how does the proposed framework perform or degrade when these assumptions are explicitly violated? Have you considered multi-positive InfoNCE adaptations to better align the theory with actual noisy biomedical data?

3. How sensitive is the overall model performance to the choice of the weight parameters ($\lambda$) for the different terms in the composite loss function? Could you provide a brief ablation study varying these weights?

**Limitations:**

yes

**Strengths And Weaknesses:**

Strengths：

1. The paper proposes a novel framework that uses single-cell gene expression as a bridge modality to transitively align histopathology images and text, addressing the scarcity of paired annotations.

2. The methodology is supported by a theoretical analysis that establishes an upper bound on the unobserved InfoNCE loss based on the separation margins of the observed modality pairs.

3. The approach demonstrates practical utility in relevant biomedical tasks, showing measurable improvements in zero-shot cell type prediction and cross-modal retrieval under low-data regimes.

4. The manuscript is clearly written and provides useful practical insights through data harmonization diagnostics, while honestly reporting its biological and methodological limitations.


Weaknesses：

1. The assumptions supporting the core theory (Lemma 3.1), such as uniform margins and single positive pairs, are overly strict. In real-world biomedical data, the coexistence of multi-positives and label ambiguity is the norm, which severely limits the strict applicability of this theoretical bound to actual noisy data.

2.  In the zero-shot classification and low-data experiments, the paper directly compares its method with external models (e.g., CONCH and PLIP) pretrained on completely different corpora. This lack of strictly controlled internal baselines makes it difficult to determine whether the performance gains stem from the proposed "trimodal architecture" itself, or simply from the inclusion of specific single-cell gene expression datasets.

3.  The paper does not provide ablation studies on the weight parameters ($\lambda$) for the different terms in the composite loss function. Consequently, it is impossible to evaluate whether the model's performance is highly sensitive to the balance between different modalities, which weakens the demonstration of the method's robustness under varying hyperparameter settings.

---

> ### Author Rebuttal · Authors · 2026-03-31
>
> We thank the reviewer for the thoughtful and constructive review. In response, we prepared additional controlled baselines, a loss-weight ablation, and clearer framing of the theory's scope.
>
> **RE Q1|W2: "Lack of strictly controlled internal baselines makes it difficult to determine whether gains stem from the trimodal architecture or the inclusion of gene expression datasets."**
>
> We appreciate this great idea. We implemented two new controlled baselines and compared to our model on the previous benchmark, as well as two newly added benchmarks for cell type prediction.
>
> | Model | PathoCell (13 classes) | PanNuke (4 classes) | Lizard (3 classes) |
> | :---- | :---- | :---- | :---- |
> | PLIP | 0.488 | 0.588 | 0.528 |
> | CONCH | 0.546 | 0.624 | 0.581 |
> | *Two-stage pipeline* | 0.550 | *\** | *\** |
> | *Bimodal I↔T (QUILT-1M only)* | 0.566 | *\** | *\** |
> | **Our model** | **0.630** | **0.689** | **0.764** |
>
> (\*pending processing; to be included in revision)
>
> The two-stage baseline implements a strengthened version of the reviewer's suggestion, trained on the same data as our model:
>
> * Step 1: UNI2 features+decoder trained on HEST-1K to predict *17,851 genes*
> * Step 2: A frozen Geneformer with a trained classification head, mapping to the benchmark classes.
>
> We will further optimize this baseline with published H\&E→ST models prior to reporting the full table in the revised manuscript.
>
> **RE W1|Q2: Assumptions supporting Lemma 3.1 (uniform margins, single positive pairs) are overly strict. How does the framework perform when these assumptions are violated?**
>
> We thank the reviewer for these thoughtful concerns.
>
> We see theory and experiments as complementary: The theory identifies the mechanism and failure modes. In practice, however, the method continues to perform well even when these assumptions are violated, suggesting that the core phenomenon likely holds under weaker conditions. While one could posit alternative, data-dependent assumptions, these may be even less realistic in biological settings, where datasets are highly diverse and noisy. Our theoretical analysis is therefore not intended to provide a learning-theoretic guarantee, but rather to clarify the geometry of the alignment process and to identify the regimes in which the method may fail.
>
> We consider the suggestion for a multi-positive InfoNCE extension, as a valuable direction and will discuss it as future work in the revision. Deriving and properly validating a new theoretical result for the multi-positive transitive setting is beyond the scope of this rebuttal phase.
> In our specific use case, single-cell resolution datasets (Visium HD, CosMx) are becoming increasingly available, enabling modeling where each data point reflects a single cell, which closely matches the lemma's single-positive assumption.
>
> **RE W3|Q3: No ablation studies on the weight parameters for the composite loss function. How sensitive is the model to these weights?**
>
> We appreciate this request, as it aligns with our goal to thoroughly characterize transitive representation learning, from which this hyperparameter naturally emerges. In a new ablation, we now **trained three models, with varying effective λ ratios** between the I↔G and T↔G training signals, by adjusting dataset composition while fixing the training budget.
>
> | Config | Cell type annotation (macro AUROC) |
> | :---- | :---- |
> | Balanced (1:1) | 0.624 |
> | Image-heavy (1:2) | 0.620 |
> | Text-heavy (2:1) | 0.605 |
>
> We will provide an expanded version of this analysis (multiple seeds, 2 more lambdas) as well as additional ablations on encoder modules and freezing configurations in the revised manuscript.
>
> We hope these additions resolve the main concerns regarding (i) isolating the contribution of the trimodal architecture via a bimodal I↔T baseline that disentangles the data component, (ii) robustness of the method to loss weighting and hyperparameter choices, and (iii) the scope of Lemma 3.1 as a theoretical diagnostic rather than a literal noise model.

---

> > ### Author Rebuttal · Reviewer_rVc4 · 2026-04-03
> >
> > Thank you for the thoughtful rebuttal and for providing the additional experiments. The new controlled baselines (bimodal and two-stage) effectively address my main concern regarding the need to isolate the trimodal architecture's contribution from the data component. The loss weight ablation also provides valuable evidence of the method's robustness. While I appreciate the clarification on the scope of the theory, the new empirical results are the most significant addition. I am satisfied with the response and will maintain my current score.

---

> > > ### Author Response · Authors · 2026-04-07
> > >
> > > We thank the reviewer for their continued effort and constructive criticism. We are pleased that we were able to fully address their concerns, and remark that this dialogue has significantly strengthened the revised manuscript.

---

### Official Review · Reviewer_9wBy · 2026-03-13

**Soundness:** 2
**Presentation:** 2
**Significance:** 2
**Originality:** 2
**Overall Recommendation:** 4
**Confidence:** 4

**Summary:**

This paper proposes a trimodal contrastive learning framework for histopathology, gene expression, and text, with the goal of improving fine-grained histopathology annotation when direct supervision is scarce. The key idea is to use paired image–gene and gene–text datasets to induce image–text alignment through a shared embedding space, which the authors refer to as transitive representation learning. The paper also provides a simplified theoretical analysis of when such transfer can succeed, and evaluates the method on zero-shot cell type prediction, low-data multimodal benchmarks, and a text harmonization experiment to study shared-modality overlap.

**Compliance With Llm Reviewing Policy:**

Affirmed.

**Final Justification:**

This paper addresses an important problem in computational pathology and presents a clear trimodal framework with meaningful empirical results. My initial concerns were about limited methodological novelty, the lack of comparison to a stronger two-stage transcriptomics-based baseline, questions about robustness beyond malignant tissue, and the gap between the idealized theory and practical biological noise. The rebuttal addressed these points well: the added two-stage baseline clarified the value of the proposed formulation, the additional benign/malignant analysis strengthened the generality claim, and the new robustness analyses on text variation and harmonization made the empirical story more convincing. I still think the main novelty lies more in the integration strategy and biomedical application than in a fundamentally new learning algorithm, and some limitations remain in terms of theory realism and absolute performance. However, the rebuttal substantially improved my confidence in the paper’s soundness and significance. Overall, this changed my evaluation positively, and I increased my recommendation from 3 to 4.

**Key Questions For Authors:**

1. The paper mainly compares against pathology vision-language baselines such as CONCH and PLIP. How does the proposed trimodal image–gene–text alignment compare to the alternative pipeline of predicting transcriptomic profiles from H&E and then performing downstream cell annotation or molecular interpretation? In particular, would the proposed approach provide better annotation accuracy than such a more direct transcriptomics-based pipeline?
2. The method relies on gene expression as a bridge to cell-type text, but such annotations are often noisy and reference-dependent in practice. How robust is the proposed framework to this kind of annotation uncertainty?
3. The theoretical analysis is based on rather idealized assumptions, while most of the empirical results are shown on diseased, especially cancerous, tissues. Can the authors clarify what concrete practical insight or diagnostic the theory provides, and whether the same mechanism is expected to hold beyond strongly pathological settings?
4. In Section 4.3, how do the authors distinguish improved shared-modality overlap from a simpler prompt/style optimization effect after text harmonization?

**Limitations:**

Yes

**Strengths And Weaknesses:**

Strengths:
1. The paper addresses an important limitation in computational pathology, namely the scarcity of fine-grained cellular annotations in routine H&E data. Using transcriptomic information as an auxiliary source of biological semantics is a meaningful direction.
2. The paper presents a clear story: image–gene and gene–text paired data are used to induce image–text alignment, and the theoretical and experimental sections are reasonably aligned with this central idea.
3. The three main experiments correspond to the paper’s theoretical motivations: whether transitive transfer works at all, whether it helps in low-data regimes, and whether overlap in the shared modality matters.


Weaknesses:
1. At the technical level, the method mainly combines standard modality-specific encoders with a shared embedding space and pairwise InfoNCE losses across three modality pairs. The main contribution feels more like a reasonable multimodal reformulation than a genuinely new learning algorithm.
2. The paper mainly compares against pathology vision-language baselines such as CONCH and PLIP, which are relevant for zero-shot prompting, but it does not compare against an important and arguably stronger alternative paradigm: predicting transcriptomic profiles directly from H&E images, followed by downstream cell-type annotation or molecular interpretation using marker genes or reference-based mapping. Since these methods pursue a closely related scientific goal—recovering molecularly informed cellular semantics from routine histology—the absence of such baselines makes it difficult to assess when the proposed trimodal image–gene–text alignment is actually preferable in practice. As a result, the necessity and practical advantage of the trimodal formulation are not yet fully established.
3. Most of the main qualitative and quantitative results are presented on pathological settings such as colorectal cancer and lung cancer. Since malignant tissues often exhibit stronger and more distinctive morphological patterns, it remains unclear whether the learned alignment is equally effective on normal tissues, where visual differences can be subtler. Showing results on normal tissue would help better assess the robustness and generality of the proposed framework.
4. The method operates on patch-level image embeddings and aligns them with gene and text representations, but it does not explicitly model cell-cell spatial relationships or microenvironmental interactions. This may limit its ability to capture biologically important tissue context beyond coarse local morphology.
5. In particular, Lemma 3.1 mainly formalizes a natural geometric intuition—if image–gene and gene–text pairs are well aligned, then image–text alignment may improve as well—under simplified margin and overlap assumptions. However, the practical difficulty in this problem is precisely that the G \leftrightarrow T supervision is often imperfect: cell-type annotations derived from gene expression are frequently noisy, reference-dependent, and inconsistent across datasets, with ambiguity in marker specificity, annotation granularity, and labeling conventions. As a result, the kind of clean positive/negative separation and reliable shared-modality alignment assumed by the theory may be hard to realize in realistic biomedical settings. This makes the theoretical section feel somewhat auxiliary: it explains the intended mechanism under idealized conditions, but does not fully address whether the core transitive bridge remains trustworthy when the biological annotation pipeline itself is uncertain.

---

> ### Author Rebuttal · Authors · 2026-03-31
>
> We thank the reviewer for the thoughtful review and for recognizing the value of using transcriptomic supervision to overcome the scarcity of fine-grained histopathology annotations. Our response contains additional analyses, including (i) a rigorous two-stage baseline, (ii) assessing performance in non-malignant tissue through additional benchmarks, and (iii) testing robustness to syntactical variations in the text modality.
>
> **Re W1: “The method feels like a reasonable multimodal reformulation rather than a genuinely new learning algorithm”**
>
> While our architecture leverages established contrastive mechanisms and builds upon initial transitive alignment observations (Girdhar et al. 2023), our primary contribution is **formalizing** this into a **theoretically grounded** and **rigorously validated** framework for the **biomedical domain**. We will reinforce this point by revising our contributions section.
>
> Specifically, we analyze the conditions under which transitive transfer succeeds or fails (further details in response to ZBSb), and we demonstrate its utility through zero-shot single-cell identification in routine H\&E slides. Regarding the latter, we now evaluated our approach on two additional independent benchmarks:
>
> | Benchmark | Ours (AUROC) | Best baseline (CONCH) | Δ |
> | :---- | :---- | :---- | :---- |
> | PanNuke | 0.689 | 0.624 | \+10.4% |
> | Lizard | 0.764 | 0.581 | \+31.5% |
>
> **Re Q1|W2: Lack of comparison against a two-stage H\&E→transcriptome→cell-type annotation baseline**
>
> We trained a **two-stage baseline** (UNI2 → decoder predicting 17,851 genes on HEST-1K → Geneformer with linear head → cell type classes) and evaluated it on the PathoCell benchmark:
>
> | Model | macro AUROC |
> | :---- | :---- |
> | **Trimodal (ours)** | **0.630** |
> | Two-stage baseline  | 0.550 |
>
> **RE W3|Q3: “results on normal tissue would help better assess the robustness and generality of the proposed framework”**
>
> Following the setting in Huang et al., 2023 we now evaluated our model's cell type annotation performance on stratified benign and malignant tissue from the PanNuke dataset. We observed **higher performance in the benign** tissue (macro AUROC 0.765 vs 0.642 in malignant tissue), providing evidence that the approach is not dependent on malignancy-driven morphology.
>
> **RE W4: Not explicitly modeling cell-cell spatial relationships may limit the ability to capture biologically important tissue context**
>
> We agree that our model does not explicitly represent spatial cell-cell interactions, viewing this as a deliberate scoping decision rather than limitation. Our aim is to provide improved **fine-grained annotations that serve as high-quality inputs for downstream microenvironmental analyses**, as developed by the spatial transcriptomics community. That said, we note that spatial relationships may be partially learned through UNI2’s patch-level pretraining.
>
> **RE W5|Q2|Q3: The theory assumes clean alignment, but G↔T annotations are noisy in practice. How robust is the framework, and what practical diagnostic does the theory provide?**
>
> *Role of the theory:* Lemma 3.1 is not meant as a guarantee that transitive learning will work. It is a design diagnostic that identifies the controllable factors: alignment quality of each modality pair, and compatibility of the shared modality across datasets. Section 3.4 explicitly relaxes perfect overlap, showing degradation scales smoothly as O(√δ). Our experiments are in line with what the theory predicts (Section 4.3; Q4).
>
> *Robustness to G↔T annotation noise:* We appreciate this concern, but also note that the G↔T link in our setup is no more ambiguous than image captions in standard CLIP.  We support this notion empirically in additional analyses, showing **modest performance variation in response to prompting templates** (AUROC between 0.628 and 0.647 across four templates) and rewording of cell type labels (\<1% macro-AUROC variation across 12 synonym/domain-specific remappings).
>
> **RE Q4: In Section 4.3, how do the authors distinguish improved shared-modality overlap from a simpler prompt/style optimization effect after text harmonization?**
>
> We appreciate this critical thought and recognize the challenge of disentangling the two components.
> To gain more confidence in the proposed mechanism—improved information transfer through enhanced modality overlap—we tested it directly: We trained two models using both G↔T and T↔I data; once using uncurated, once using curated T↔I QUILT-1M texts. We evaluated these models on our G↔I benchmark (HEST-1K benchmark, Fig. 6). The model with the **curated text bridge exhibited increased performance across 9 of 10 tissue types** in the benchmark (p=0.024, Wilcoxon), providing further evidence that text harmonization strengthens the transitive bridge.
>
> We highly appreciated the reviewer's concerns and we will report the resulting findings in the revised manuscript.

---

> > ### Author Rebuttal · Reviewer_9wBy · 2026-04-02
> >
> > Thank you for the thoughtful rebuttal and for adding a two-stage baseline, additional benchmarks, and robustness analyses. These additions are helpful and do address part of my original concerns.
> >
> > Given the goal of the paper, I believe there is still an important alternative paradigm that deserves clearer discussion and, where possible, comparison: predicting gene expression / spatial transcriptomics from H&E images, and then using those predictions for downstream cell-type annotation or molecular interpretation. Since the present work, such as GHIST, Loki, and sCellST, ultimately also aims to recover fine-grained cellular semantics from histology, this is a highly relevant point of comparison. While these methods are technically different from the proposed trimodal framework, they are closely related in scientific objective. I would therefore encourage the authors to clarify more explicitly in the revision under what conditions the proposed image-gene-text framework should be preferred over this alternative route. Even if a full empirical comparison is beyond the scope of the revision, the paper would benefit from a clearer discussion of these related approaches, their differences, and the practical trade-offs involved.

---

> > > ### Author Response · Authors · 2026-04-03
> > >
> > > We thank the reviewer for the thoughtful follow-up. We agree that this intuitive baseline needs to be discussed in a dedicated subsection in the revised paper, which we will do.
> > >
> > > **When and why should our approach be preferred?**
> > >
> > > The core issue with the H&E → expression → cell type paradigm is related to the fact that histology images are underspecified in representing the transcriptome state.
> > > Therefore, in a two-step prediction scenario, the intermediary transcriptome exhibits substantial noise, which the downstream classifier ingests blindly.
> > > Our approach mitigates this issue through a shared embedding space, where the encoders jointly learn representations that are relevant and focused for downstream tasks.
> > >
> > > To answer the reviewer's question directly: Our approach is better suited for predictions at the cellular level (e.g. cell types), as opposed to the prediction of one, or few, specific genes.
> > >
> > > **Evidence**
> > >
> > > In addition to our two-stage baseline we now also include the performance of OmiCLIP. OmiCLIP's zero-shot annotation pipeline (Loki Annotate) performs zero-shot cell type annotation by scoring similarity of H&E patches against cell type marker genes.
> > >
> > > Despite testing two sets of representative marker genes, following their pipeline and recommendations, OmiCLIP's approach only recognized B cells at meaningful performance (0.646 AUROC) whereas other cell types were near random, reinforcing the importance of learning all three aspects jointly: image, expression state, and annotation.
> > >
> > >
> > > | Method                        | AUROC (PathoCell; mean across 13 cell types) |
> > > |------------------------------|-------|
> > > | Trimodal (ours)              | 0.630 |
> > > | Two-stage baseline (UNI2→GF) | 0.550 |
> > > | CONCH                        | 0.545 |
> > > | OmiCLIP (short marker list)  | 0.488 |
> > > | OmiCLIP (expanded list)      | 0.480 |
> > >
> > >
> > > The revised manuscript will include this and the two-stage baseline, scored for all three benchmarks (PathoCell, PanNuke, Lizard), alongside our guiding interpretations.

---

### Official Review · Reviewer_2iDN · 2026-03-13

**Soundness:** 3
**Presentation:** 3
**Significance:** 2
**Originality:** 2
**Overall Recommendation:** 4
**Confidence:** 3

**Summary:**

The paper proposes a trimodal contrastive learning framework for zero-shot annotation of histopathology images. The core idea is to use gene expression as a bridge modality, training on disjoint paired datasets of image-gene expression (spatial transcriptomics) and gene expression-text (single-cell annotations) to indirectly learn image-text alignment. The shared embedding space is learned via InfoNCE-based contrastive objectives and supports zero-shot cell type prediction via natural language queries. The authors also provide a theoretical bound justifying when transitive transfer should succeed, and report improvements over two pathology vision-language baselines.

**Compliance With Llm Reviewing Policy:**

Affirmed.

**Final Justification:**

The rebuttal addressed all main concerns, including multi-seed variance, the expanded QUILT-1M analysis, and the two-stage baseline that isolates the learning strategy from the data. Soundness and presentation are good, and the theory is clean and well tied to the experiments. The main limitations are on originality and significance: the algorithmic contribution over ImageBind-style hub alignment is modest, and absolute cell-typing performance remains moderate. The paper makes a solid empirical contribution to an important problem in computational pathology and I expect others to build on the data integration strategy, but these limitations keep this at a weak accept.

**Key Questions For Authors:**

1. The main results in Table 2 are reported from a single training run. Can the authors provide variance estimates across multiple seeds? Given that only the projection heads and text encoder are trained for 4 epochs with frozen backbones, this should be computationally feasible and would meaningfully strengthen confidence in the reported gains, particularly for cell types where the margin over baselines is small.
2. The transitive transfer relies critically on Geneformer producing a consistent embedding space across both HEST-1K (Visium spatial transcriptomics) and CellWhisperer (single-cell RNA-seq), despite these datasets coming from different measurement platforms with known technical differences in sequencing depth and cell resolution. Could the authors empirically verify this assumption by embedding gene expression profiles from both datasets through the frozen Geneformer encoder and visualizing their overlap via UMAP or t-SNE? Substantially disjoint distributions would suggest that the bridge modality is not truly shared, which would be an important qualifier on the main results and would help ground the theoretical analysis in the actual data used for training.

**Limitations:**

Yes, regarding limitations. The authors discuss patch-level resolution constraints, encoder choices, and evaluation scope honestly. On societal impact, the statement that no specific concerns need highlighting is reasonable given the early-stage nature of the work, though a brief note on risks of deploying unreliable cell type predictions in clinical contexts would have been appropriate.

**Strengths And Weaknesses:**

The paper tackles a real and well-motivated problem in computational pathology and the proposed solution is clean and well-executed. The theory is nicely tied to the experimental design, and the gains over baselines on cell type prediction are convincing, especially for immune populations. The main concerns are the lack of variance estimates in the main results, the perfect overlap assumption in the theoretical analysis which does not hold in practice, and somewhat limited absolute performance on several cell types. A minor presentation concern is that an interesting negative result gets only brief mention in the main text.


**Soundness**

Strengths:

1. Theoretical motivation: Lemma 3.1 provides a formal bound linking the unobserved image-text loss to the margins on the observed training pairs, giving a principled justification for why the approach should work.

2. Theory-to-experiment alignment: The three experiments directly probe the three main claims from the theoretical analysis in sequence: that transitive learning induces image-text alignment, that it complements direct supervision, and that shared modality overlap is necessary. This is a nice feature of the paper's structure.

Weaknesses:

3. Lack of variance estimates: The main results in Table 2 report only mean AUROC across samples without error bars or variance across training runs. Given that the heavy backbones are frozen and only projection heads and the text encoder are trained for 4 epochs, running multiple seeds would be computationally cheap and would meaningfully strengthen the robustness of the reported results.

4. Theory-practice gap: The theoretical bound assumes perfect overlap in the shared gene expression modality across datasets, which the authors themselves acknowledge is rarely true. The LLM-based text harmonization partially addresses this but does not close the gap, leaving the theoretical analysis somewhat disconnected from the practical setting.

**Presentation**

Strengths:

5. Clarity of mechanism: The trimodal objective is well explained and the geometric visualizations in Figures 2 and 3 effectively build intuition for how the shared embedding space is shaped during training.

Weaknesses:

6. Underreported negative result: The finding that adding QUILT-1M image-text data hurts performance on the cell type benchmark is mentioned in the main text in a single sentence and analyzed only in Appendix B. This is not a fatal issue, but the result is genuinely interesting as it suggests that task-relevance of auxiliary data matters at least as much as its availability. A bit more discussion in the main text would have made the paper's narrative more complete and balanced.

**Significance**

Strengths:

7. Relevant problem: The combination of cheap but coarsely annotated histopathology images with expensive but richly labeled gene expression data is a genuine bottleneck, and the proposed approach offers a practical route to bridging these data sources.
Low-data regime performance: The consistent gains over bimodal baselines when target data is heavily subsampled (down to 1/64 or 1/512) is a practically useful result given how often biomedical annotation pipelines are data-limited.

Weaknesses:

8. Absolute performance: The mean AUROC of 0.630 is modest, and some cell types such as NK cells (0.457) and Adipocytes (0.473) perform near or slightly below random chance. The authors provide reasonable explanations for these cases, but they do highlight that the method is not yet ready for reliable clinical use.

**Originality**

Strengths:

10. Data integration strategy: The specific combination of spatial transcriptomics and single-cell atlases as a route to grounding histopathology interpretation in molecular concepts is a sensible and well-executed idea that others are likely to build on.

Weaknesses:

11. Algorithmic novelty: The hub-based multimodal alignment concept builds directly on prior work like ImageBind, as the authors acknowledge. The novelty is primarily in the biological application and the data combination strategy rather than in the machine learning methodology itself.

---

> ### Author Rebuttal · Authors · 2026-03-31
>
> We thank the reviewer for the thoughtful and encouraging review, and for highlighting both the practical relevance of the problem and the close alignment between the theory and experiments. In response to the concerns, we will strengthen the paper along four main axes: (i) reporting variance across seeds, (ii) promoting the QUILT-1M negative result to the main text, and (iii) clarifying the scope and current limitations of the cell-typing benchmark.
>
> **RE W3: Table 2 reports only a single training run. Running multiple seeds would strengthen confidence.**
>
> We agree this is an important addition to assess robustness. We have trained two additional models, confirming stable performance advantages over baselines in additional benchmarks.
>
> | Benchmark | *Ours seed=0* | Ours seed=1 | Ours seed=2 | Ours mean ± std | PLIP | CONCH |
> | :---- | :---- | :---- | :---- | :---- | :---- | :---- |
> | ***PathoCell*** | 0.630 | 0.614 | 0.620 | **0.621 ± 0.006** | 0.468 | 0.507 |
> | **Lizard** | 0.764 | 0.732 | 0.752 | **0.742 ± 0.010** | 0.528 | 0.581 |
> | **PanNuke** | 0.689 | 0.687 | 0.686 | **0.687 ± 0.001** | 0.588 | 0.624 |
>
> We will include these benchmarks in the revision.
>
> **RE W4|Q2: The theoretical bound assumes perfect overlap in the shared modality across disparate datasets, which is rarely true and leaves the theory disconnected from the practical data.**
>
> As detailed in Section 3.4, our theory explicitly accommodates imperfect overlap, demonstrating that modality mismatch introduces a predictable additive slack of $O(\\sqrt{\\delta})$ rather than breaking the transitive bridge entirely. Indeed, geometric separation of modalities in CLIP-style models is the norm rather than a barrier to semantic transfer (Liang et al., 2022).
>
> While transcriptomics technologies exhibit platform-specific batch effects (as seen in UMAP), our theory suggests that the bound is strengthened through shared-modality harmonization, which we demonstrate empirically (Section 4.3; W6).
>
> We will clarify these points in the revised manuscript.
>
> **RE W6: The finding that adding QUILT-1M image-text data hurts performance is very interesting and deserves more than a brief mention.**
>
> We appreciate this feedback,  which has prompted us to expand this analysis. We trained a full trimodal model incorporating the LLM-harmonized QUILT-1M dataset from Section 4.3:
>
> | Model | PathoCell (macro AUROC) | Δ vs bridge |
> | :---- | :---- | :---- |
> | Bimodal bridge | 0.630 | — |
> | Trimodal \+ QUILT-1M | 0.609 | −3.3% |
> | **Trimodal \+ harmonized QUILT-1M** | **0.645** | **\+2.4%** |
>
> This reinforces our points in Section 3.4 ( "Effect of imperfect overlap in $\\mathcal{G}$" and "Implications for training on all three edges"). We will promote these findings in the main text.
>
> **RE W8:  Mean AUROC of 0.630 is modest. The method is not ready for reliable clinical use.**
>
> We agree with the reviewer that the method is not ready for clinical deployment. We will reinforce this in the Limitations section.
>
> However, we emphasize that zero-shot, **fine-grained classification of 13 distinct cell types from routine H\&E patches is an exceptionally difficult task that has not been previously demonstrated**. To contextualize our absolute performance, we evaluated our model on two additional, widely recognized benchmarks using coarser-grained cell types (see RE W3).
>
> We will include these benchmarks in the revision.
>
> **RE W11: The hub-based alignment concept builds directly on prior work like ImageBind; novelty is mostly in the biological application.**
>
> Girdhar et alia’s seminal work (2023, CVPR) empirically described “emergent” transitive alignment. Our work recognizes the relevance of this mechanism and develops it into a **theoretically grounded learning framework that facilitates impactful use**.
>
> We further underline the relevance of our contribution through another baseline that disentangles the effect of training data from our learning strategy. We trained a two-stage pipeline (H\&E → transcriptome → cell-type) on the same data as our model and assessed its performance on PathoCell (AUROC 0.550 vs. **0.630 (ours)**). This demonstrates that the learning strategy, not the data, drives the gains.
> We will refine our contributions section to clarify our novelty over previous work.
>
> **Our work addresses an unsolved problem in computational pathology**: obtaining fine-grained cell-type annotations from routine H\&E without task-specific labeled data. The revised analyses confirm substantial gains over existing VLMs and the two-stage alternative across three independent benchmarks, and **provide actionable guidance for when transitive learning succeeds**. We believe this principled bridge between the single-cell transcriptomics ecosystem and histopathology interpretation will be broadly useful to the community.

---

> > ### Author Rebuttal · Reviewer_2iDN · 2026-04-04
> >
> > Thank you for the thorough rebuttal. The multi-seed results (W3) convincingly establish robustness, and the harmonized QUILT-1M experiment (W6) is a valuable addition that directly connects the text alignment analysis to the main result. The two-stage pipeline baseline (W11) also helpfully disentangles the contribution of the learning strategy from the data. The core concerns have been adequately addressed and I will maintain my current score.

---

> > > ### Author Response · Authors · 2026-04-07
> > >
> > > We thank the reviewer for their feedback and appreciate that we resolved all their concerns.

---

### Decision · Program_Chairs · 2026-04-30

**Decision:**

Accept (regular)

**Comment:**

The paper proposes acontrastive learning framework for zero-shot annotation of histopathology images. The core idea is to use gene expression as a bridge modality between images and text, training on independent paired datasets of image-gene expression and gene expression-text to indirectly learn image-text alignment. Strengths of the paper include the clear motivation from a real histopathological problem, the theoretical investigation and the link of the experiments to the theory. Questions arose from the lack of variability quantification, the suitability of spatial transcriptomics as a bridge modality, overly strict theoretical assumptions and lack of exploration of alternative model layouts. Overall, the rebuttal by the authors adressed many of the points to the satisfaction of the reviewers, such that three weak accept and one weak reject seem to argue in favor of **accept**.